# A matching method for elderly care service personnel with multiple types of service expectations

Chao Yu*, Tianxiang Gao*

School of Management, Shenyang University of Technology, Shenyang, China

* 493782958@qq.com (TG); yuchao_neu@163.com (CY)

## Abstract

With the rapid growth of the global aging population, the problem of providing for the elderly has become increasingly prominent. As a new model of providing for the elderly in China, home-based care has attracted more and more attention from all walks of life. The research on how to realize reasonable and effective matching between elderly care service personnel and the elderly under the home care model deserves attention and has important practical significance. In this paper, a matching method for elderly care service personnel considering multi-type service expectations is proposed. This approach involves first obtaining the actual values of the expectation indexes for each other, as well as the expectation requirements of the elderly and the service personnel. Next, we compute the satisfaction of the elderly and the service personnel for each other based on the elastic service expectation type, and finally, we make a decision about whom to choose based on the inelastic service expectation indexes. On this basis, the situation is considered that there are sufficient and insufficient elderly care service personnel, the two-sided matching models are constructed, respectively, and the optimal matching results are obtained by solving the models. Finally, an example is given to illustrate the feasibility and effectiveness of the proposed method.

## 1. Introduction

One of the most significant social shifts of the twenty-first century is the aging of the population, which will affect most facets of society, such as the labor market, the need for services, and family dynamics. As per the United Nations Department of Economic and Social Affairs' World Population Prospects (2019) [1], a society is deemed to be aging when the percentage of its population that is 65 years of age or older surpasses 7%. Global population aging is a result of low fertility rates, rising life expectancy, and labor force migration. The number of people 65 and older has increased sharply in recent years. The number of adults 65 and over crossed 700 million in 2020 for the first time, coming in at 723.484 million [2]. The number of people 65 and older worldwide is predicted to reach 1.5 billion by 2050. Furthermore, according to a study on aging in China, the country's senior population—those over 65—will number 191 million in 2020, rise to 300 million in 2030, and reach 420 million in 2050 [3]. The State

**Data Availability Statement:** All relevant data are within the paper.

**Funding:** This work was supported in part by the Liaoning Provincial Social Science Planning Fund

[L21CGL021 to CY]. No additional external funding was received for this study. The funder had no role in study design, data collection and analysis, decision to publish, or preparation of the manuscript.

**Competing interests:** The authors have declared that no competing interests exist.

Council's seventh national population census revealed that the percentage of Chinese citizens 60 years of age and older was 18.7%, while the percentage of those 65 and older was 13.5% [4]. China has entered an aging society since 2001 [5], based on this standard. This percentage is predicted to increase by 247% between 2015 and 2050, while the percentage of adults over 80 is predicted to increase by 522% during the same time frame [6]. The issue of aging has become more pressing for all facets of society as a result of the aging population's rapid growth, surpassing that of the first developed nations. China's population is aging at a rate that has been steadily rising as well [7]. As a result, the need for elderly care services has grown.

At the moment, there are five main categories into which the global mainstream aging models can be divided: the Continuing Care Retirement Community (CCRC) aging community model, which is best represented by Sun City in the United States [8], family-based aging, which is represented by Singapore and Japan [9], the transnational aging model, which is represented by the Nordic countries of Sweden, Norway, Finland, Denmark, and Iceland [10], the time-banking aging model, which is represented by Switzerland [11], and the intergenerational learning center aging model, which works with nursing homes and kindergartens [12]. China's traditional pension model is based primarily on family pension as an addition to institutional pension due to differences in geography, beliefs, customs, and habits. However, China has a higher population density than the rest of the world, and the one-child policy, which was implemented between 1982 and 2016, resulted in a decrease in the number of young people in the generation that made up the base, the emergence of the phenomenon of two one-child children who married and had to provide for at least four elderly people, and a gradually growing burden on old age. In addition, China's aging population necessitates a wide range of services, yet the country lacks adequate resources for the elderly, so applying other nations' aged care models directly to China's own situation is not wholly appropriate. China's distinct political structure, fundamental national circumstances, historical and cultural heritage, and other factors contribute to the peculiarities of the aging issue and its range of remedies. In China, home care, community care, and institutional care are the three categories of elder care [13]. Home care is more popular among the elderly due to the high cost and lack of resources associated with community and institutional care; major Chinese cities have developed a pattern of elderly care systems that place home care as the primary, community care as a supplement, and institutional care as a supplement [14]. With "more dispersed personnel, door-to-door service" characteristics, home care refers to the family as the center, depending on professional services, to provide socialized services to the elderly living at home in order to solve the challenges of everyday life [15]. It should be noted that the issue of staff matching for in-home senior care services involves finding senior care providers who are suitable for the elderly. This matching process takes into account factors such as social dynamics, time constraints, and individual characteristics in addition to pure profit. From an operational standpoint, the key to guaranteeing that senior citizens receive high-quality services is the appropriate and optimal matching of resources for elderly service personnel [16].

As the need for home care services rises, the market for smart home care services is predicted to explode in the ensuing decades. But there are other obstacles in the way of smart home care services becoming widely used. The federal government of China is accelerating the provision of home care services to the country's old population due to the elderly's physical and psychological vulnerabilities as well as the lack of care services available to them [17]. As a reflection of the idea of integrated healthy aging, home care services are basically a combination of health and aged care in the current Chinese setting [18]. Stated differently, it is frequently regarded as a home-based senior care service that offers life care, medical attention, medical prevention, rehabilitation training, and other services to senior citizens who age in place [19]. The home-based elderly care service matching problem differs from the traditional

domestic service matching problem [20] in three ways: first, the service object is exclusive to the elderly; second, door-to-door service personnel must possess a wide range of service skills; and third, the service items are more specific and in-depth. According to the literature [20], problems of low matching efficiency and low satisfaction among the elderly may arise when the traditional matching method is used. The various types of service expectations that the elderly and the service personnel have, such as the inelastic expectations regarding service content and skill level and the elastic expectations regarding age and salary, must therefore be taken into account in order to fully utilize the service resources and achieve the best matching effect. Additionally, there may be an imbalance between the supply and demand of elderly service resources in actual life. The primary innovation section of the article, however, also highlights the importance of paying attention to how to address the issue of matching service personnel with multi-type service expectations and adapting service resources based on various circumstances based on the smart elderly people platform [21]. For these reasons, a targeted matching approach for various scenarios is required. In light of this, this study suggests a matching approach for senior service employees that takes into account various service expectations based on the potential mismatch between the supply and demand of senior service resources. The method's main idea is to: (1) look into the various kinds of expectations that the elderly and service personnel have of each other; (2) find the elderly and service personnel's actual values for each other's expectation indexes; and (3) compute the elderly's satisfaction with the service personnel as well as the service personnel's satisfaction with the elderly. Based on this, the matching model is built using the two real-world scenarios of having enough and not enough elderly service personnel. The model's solution method is presented in a targeted manner, and the matching program for senior service personnel is ultimately decided by using the matching model's solution.

## 2. Related work

### 2.1 Elderly services issues with multiple types of service expectations

The topic of aging with regard to service expectations of older persons and service providers is complex, according to research findings from the past. For example, the application of modularity in service settings—more especially, in the delivery of care and services to older adults who live independently—was illustrated by De Blok C. et al. [22]. In order to personalize ICT to reduce social isolation and loneliness in older persons, Thangavel et al. [23] stressed the significance of discussing and managing social isolation and loneliness as distinct, but connected, concepts. In order to assist in putting knowledge into reality, Dong Y and Dong H [24] designed a set of design tools based on the resources and empowerment patterns of older people. They saw these individuals as positive aspects and resources. A system built by Ianculescu M. et al. [25] was intended to give elderly individuals in smart settings set up at home personalized non-invasive remote monitoring, health evaluation, and help. It has artificial intelligence-based multivariate predictive algorithms that track the patient's location, activities, and vital signs over time and send out notifications and reminders. Dong and associates [26] By tailoring appropriate exercise programs according to the health state of the older individuals, medical-physical integration of community-based aging in place enhances the health and quality of life of older persons and prevents geriatric illnesses. The strategic goal of Thailand as a medical center and retirement destination raised awareness of the need to improve patient care and medical standards. Khurana et al. [27] explored and analyzed the fundamentals of geriatric care, especially long-term care and the application of multidisciplinary integration in the management of geriatric care services to enhance the quality management of geriatric care and services. In their research, Santoso and Redmond [28] proposed that, by using the concept of

smart aging, wireless indoor positioning systems can help the advancement of home care systems to handle the issue of fall monitoring for older persons with visual impairments. Majumder et al.'s [29] thorough assessment of smart home healthcare technology was suggested using data on older individuals' general health that was gathered from a distance. In order to increase older persons' satisfaction, Alexandri and Tsirintani [30] integrated the personalized approach with the continual enhancement of quality indicators management skills to examine the social care and health domains of smart telemedicine utility. Szabó [31] analyzes the choice of aging models under smart information security systems by microsimulation. Through a case study of long-term care for senior citizens, De Blok et al. [32] offer insights into how customization and personalization are implemented. Personalization enhances customization by adjusting supply to demand. According to Schoen et al. [33], certain nations must enhance their aged care services by building a workforce of trained caregivers and easily accessible care facilities to facilitate a smooth transition from community-based, aging-in-place care to institutional care. Based on the Case-Based Reasoning (CBR) approach, Tang et al. [34] offer extremely accurate and responsive services to the elderly and assist the caregivers in making personalized changes to the implemented care plans. In order to provide users with more accurate tailored senior care service recommendations, Zhang et al. [35] suggested a recommendation approach based on user preferences and inter-user trust relationships. The accuracy of the recommendations is increased by the recommendation system that takes into account user preferences and relationships of trust. In order to create a matching service recommendation system, Tserpes et al. [36] employed a mapping between customer evaluations of service quality and the mapping link between quality characteristics of supplier reputation. Recommendation algorithms in line with user requirements and automatically running multivariate analytics services chosen by users within a big data platform were presented by Ku et al. [37]. Ling et al.'s [38] ontology-based social network service recommendation system was developed by enhancing the word frequency inverse document frequency method and the semantic similarity algorithm.

## 2.2 Two-sided matching of older persons with service personnel

According to the findings of earlier research, numerous studies have been conducted on the matching problem of senior service members, and some of these studies have also looked at the bilateral matching problem of persons. For instance, in their study of male-female marital matching and college admissions, Gale and Shapley [39] initially developed the idea of stable matching and provided a delayed acceptance algorithm for producing stable matches. After studying the issue of bilateral matching between interns and hospitals that provided incomplete preference ordering, Irving et al. [40] proposed a linear time algorithm that was used in numerous large-scale matching projects by analyzing the strong stable matching problem. Kojima et al. [41] developed a decision support system utilizing a stable matching algorithm and hierarchical analysis to address the issue of matching army jobs and personnel types. Chen and Song [42] examined the bilateral matching problem between banks and enterprises in the financial credit market and found that the likelihood of matching is higher when banks and enterprises are more willing to share data and expertise and are located closer to one another geographically. In order to maximize the overall utility of ships and cargoes, Peng et al. [43] addressed the matching problem between ships and cargoes in the dry bulk shipping market. They also developed a price game mechanism based on the Gale-Shapley algorithm and looked into three different scenarios: shipper-dominated, carrier-dominated, and equilibrium markets. The findings of this study indicated that weaker players may benefit more from using the price game mechanism even when they are in a disadvantageous position. In his research, Westkamp [44] looks at the German university admissions system from the

standpoint of stable matching processes and complex conditions. He puts forth two conjectures that ensure the existence of a stable and optimal student matching mechanism and demonstrates the mechanism's weak Pareto superiority over equilibrium results generated by alternative mechanisms. Marx and Schlotter [45] investigate two variants of the classical stable marriage problem: the hospital-intern matching problem and its extension, which permits intern pairs with conjugal relationships to provide a joint preference ordering of hospitals. They also provide a local search algorithm and a stochastic fixed-parameter easy-to-process algorithm for finding the greatest number of matches with conjugal relationships. A robust bilateral satisfaction matching model for carpooling based on fuzzy linguistic information processing was presented by Zhao et al. [46]. Wang et al. [47] proposed a model that combines psychological characteristics and human rational behaviors for determining the most satisfactory results of agents on both sides of the matching problem. Yu and Xu [48] designed an intuitionistic fuzzy bilateral matching model based on intuitionistic fuzzy with regard to its evaluation index, the human-job matching problem can be successfully resolved using the Choquet integral aggregation operator. A multi-granularity probabilistic linguistic bilateral matching method based on the peer effect is presented by Wang and Li [49]. It chooses the best elderly care model based on the elderly's family situation; however, other elderly care models must be chosen based on the features of the services that are available to the elderly. Zhu [50] examined the development of the Chinese university admissions process from the standpoint of the design of the mechanism. The study demonstrated that the simple sequential dictatorship mechanism eliminates reasonable envy and is manipulation-proof and Pareto effective, according to a review of theoretical and experimental mechanism design literature. This suggests that it is reasonable to switch from a sequential to a parallel selection mechanism for university admission in recent years in China. Yalçındağ et al. [51] suggested a data-driven method to calculate the caregivers' trip time in the allocation problem, taking into account the possibility that the patients' and their families' clinical conditions could have an impact on both the caregivers' travel and service times. In his discussion of the need for intelligent technology in the independent lives of the elderly, Zhou J. [52] draws on the field's experience with intelligent technology to determine how best to foster humanized interactions among the elderly. Liu et al. [53] to investigate the relationship between voice assistant feedback and information style. The findings demonstrate the benefits of visual-auditory bimodality in social interactions, as well as in terms of acceptance, satisfaction, and perceived joy. Wen and Chen's [54] research focused on improving the way that community-dwelling seniors receive emotional support services and handle their numerous service requests. Sun et al. [55] examined technologies for remote therapist participation in home- and community-based rehabilitation.

## 3. Problem description

In this work, we primarily address a Home Care Service Platform (HCSP) designed to help the elderly meet the needs of elderly care service personnel. Every elder in this kind of situation who requires assistance with aging is paired with an elderly care service personnel who offer consistent and long-term care. These are in addition to the seven indicators of age, salary, work experience, education level, gender, job content, and qualification certificate that an elder considers when choosing elderly care service personnel. Elderly care service personnel closely monitor four indexes: salary, welfare, work form, and the situation of the elderly. The elasticity index says that when the actual value of the index does not meet expectations, the old people will not reject the service personnel directly, but will reduce their satisfaction with the service personnel. The inelastic index means that when the actual value of the index does not meet the expected requirements, the elderly will refuse to choose the service personnel. We

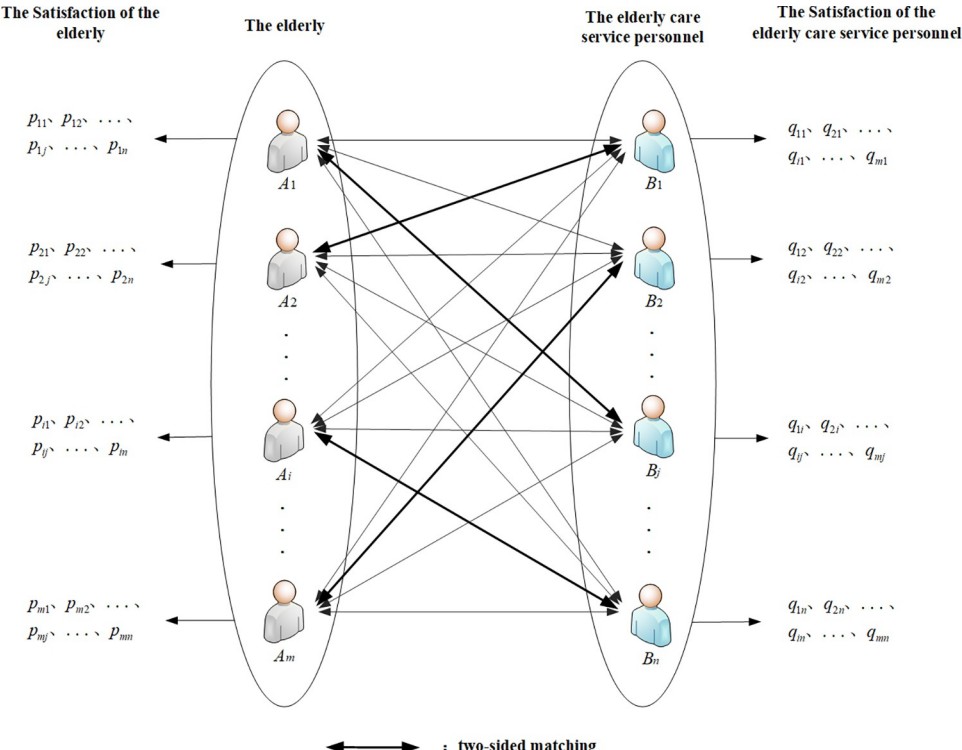

**Fig 1. The two-sided matching between the elderly and elderly care service personnel.**

should make every effort to match one elderly care service personnel with every elderly user on the platform in order to meet the needs of both parties, as there is currently an imbalance between the supply and demand of elderly care service personnel. When there are not enough resources available for services, the elderly should be served first by matching service providers with urgent needs on the platform. The two-sided matching issue between elderly and elderly care service personnel is depicted in Fig 1. In Fig 1, the thin arrow indicates that both the elderly and the service provider are satisfied with each other; the thick arrow indicates that both are matched, that is, satisfy each other's requirements and satisfy the highest overall satisfaction.

The sets and quantities involved in the elderly care service personnel matching problem when taking multi-type service expectations into consideration are listed below.

- $A = \{A_1, A_2, \ldots, A_i, \ldots, A_m\}$ represents a group of $m$ elders, where $A_i$ denotes the $i$th elder, $i = 1, 2, \ldots, m$

- $B = \{B_1, B_2, \ldots, B_j, \ldots, B_n\}$ represents a group of $n$ elderly care service personnel, where $B_j$ represents the $j$th elderly care service personnel, $j = 1, 2, \ldots, n$.

- $U = \{U_1, U_2, U_3, U_4, U_5, U_6, U_7\}$ represents the set of indicators that the elderly are concerned with when choosing elderly care service personnels, where $U_1$ represents age, $U_2$ represents salary, $U_3$ represents work experience, $U_4$ represents education level, $U_5$ represents gender, and $U_6$ represents job content, $U_7$ stands for qualification certificate.

- $E = \{E_1, E_2, E_3, E_4\}$ represents the set of indicators considered by elderly care service personnels in $E_4$ selecting elders, where $E_1$ represents salary, $E_2$ represents welfare, $E_3$ represents forms of work and represents the situation of the elderly.

- $C^i = \{c_1^i, c_2^i, c_3^i, c_4^i, c_5^i, c_6^i, c_7^i\}$ represents the set of expected values given by the $i$th elder when selecting an elderly care service personnel. Where $c_1^i$ represents the $i$ of the elderly to choose the elderly care service personnel to take into account the value of indicator $U_1$ range; $c_2^i, c_3^i, c_4^i$ represents the value of indicator $U_2$, $U_3$, $U_4$ taken into account by the $i$th elder when selecting an elderly care service personnel; $c_5^i, c_6^i, c_7^i$ represents the indicative matrix of the value of indicator $U_5$, $U_6$, $U_7$ taken into account the $i$th elder when selecting an elderly care service personnel.

- $c_1^i = [c_{1min}^i, c_{1max}^i]$ represents the interval of desired values for indicator $U_1$ when the $i$th elder chooses an elderly care service personnel, where $c_{1min}^i$ denotes the lower limit of the interval in which the $i$th elder's desired age for an elderly care service personnel is taken, $c_{1max}^i$ denotes the upper limit of the interval in which the $i$th elder's desired age for an elderly care service personnel is taken.

- $c_5^i = [\,sex_1^i \quad sex_2^i\,]$ represents the indicator matrix of the expected value of the indicator $U_5$, $sex_y^i \in \{0, 1\}$, $y = 1,2$ for the $i$th elder choosing an elderly care service personnel. When $sex_1^i = 1$ is used, it means that the desired gender indicator for the elderly care service personnel for the $i$th elder is "male"; when $sex_2^i = 1$ is used, it means that the desired gender indicator for the elderly care service personnel for the $i$th elder is "female".

- $c_6^i = [\,l_1^i \quad l_2^i \quad l_3^i \quad l_4^i \quad l_5^i \quad l_6^i \quad l_7^i\,]$ represents the indicator matrix of the value of the expectation for the indicator $U_6$, when the $i$th elder chooses the elderly care service personnel, $l_s^i \in \{0, 1\}$, $s = 1,2,\ldots,7$. Where $l_s^i$ represents the indicator information of the expectation of the $s$th item of the work content indicator for the $i$th elder choosing the elderly care service personnel, and if the $s$th item is needed, then it is taken as $l_s^i = 1$, otherwise it is taken as $l_s^i = 0$. Here, the seven items included in the work content indicator $U_6$ are: "Help with chatting", "Help with meals", "Help with cleaning", "Help with bathing", "Help with mobility", "Help with medical care" and "Help with personal care".

- $c_7^i = [\,o_1^i \quad o_2^i \quad o_3^i \quad o_4^i \quad o_5^i \quad o_6^i\,]$ represents the matrix of indications of the desired value of indicator $U_7$ for the $i$th elder's choice of elderly care service personnel, $o_t^i \in \{0, 1, 2, 3, 4, 5\}$, $t = 1,2,\ldots,6$. Where $o_t^i$ represents the information on the expectation of the $t$th component of the qualification indicator $U_7$ for the $i$th elder choosing an elderly care service personnel. The six credentials included in the credential indicator $U_7$ are: "Domestic helper certificate", "Houseworker certificate", "Home care worker certificate", "Elderly Caregiver Certificate", "Dietitian Certificate", and "Chef Certificate".

- $D^j = \{d_1^j, d_2^j, d_3^j, d_4^j, d_5^j, d_6^j, d_7^j\}$ represents the set of actual values taken by the $j$th elderly care service personnel for each desired indicator for the elderly. Where $d_1^j, d_2^j, d_3^j, d_4^j$ denotes the actual values of the $j$th elderly care service personnel for the indicators $U_1$, $U_2$, $U_3$, $U_4$; $d_5^j, d_6^j, d_7^j$ denotes the actual values of the $j$th elderly care service personnel for the indicators $U_5$, $U_6$, $U_7$ the indication matrix.

- $d_5^j = [\,sex_1^j \quad sex_2^j\,]$ represents the matrix of indications of the actual values taken by the $j$th elderly care service personnel for the indicator $U_5$, $sex_y^j \in \{0, 1\}$, $y = 1,2$. When $sex_1^j = 1$ indicates that the actual gender indicator for the $j$th elderly care service personnel is "male"; when $sex_2^j = 1$ indicates that the actual gender indicator for the $j$th elderly care service personnel is "female", $sex_1^i + sex_2^i = 1$.

- $d_6^j = [\,l_1^j \quad l_2^j \quad l_3^j \quad l_4^j \quad l_5^j \quad l_6^j \quad l_7^j\,]$ represents the indication matrix of the actual value taken by the $j$th elderly care service personnel for the indicator $U_6$, $l_s^j \in \{0, 1\}$, $s = 1,2,\ldots,7$. Where

$l_s^j$ denotes the indication information of the $s$th item of the work content indicator $U_6$ for the actual application of the $j$th elderly care service personnel, and if the $s$th item is actually applied, it is taken as $l_s^j = 1$, otherwise, $l_s^j = 0$.

- $d_7^j = \begin{bmatrix} o_1^j & o_2^j & o_3^j & o_4^j & o_5^j & o_6^j \end{bmatrix}$ represents the actual matrix of indications of the values taken by the $j$th elderly care service personnel for the indicator $U_7$, $o_t^j \in \{0, 1, 2, 3, 4, 5\}$, $t = 1,2,\ldots,6$. Where $o_t^j$ represents the actual indicator information for the $t$th element of the qualification certificate for the $j$th elderly care service personnel.

- $F^j = \{f_1^j, f_2^j, f_3^j, f_4^j\}$ represents the set of expected values given by the $j$th elderly care service personnel when selecting the elderly. Where $f_1^j$ represents the value of the indicator $E_1$ considered by the $j$th elderly care service personnel in the selection of the elderly, which is the same as the value of $d_2^j$; $f_2^j$ represents the set of values of the indicator $E_2$, considered by the $j$th elderly care service personnel when selecting the elderly; $f_3^j$ and $f_4^j$ represent the matrix of indications of the values of the indicators $E_3$ and $E_4$, considered by the $j$th elderly care service personnel when selecting the elderly.

- $f_2^j = \{z_1^j, z_2^j, z_3^j, z_4^j, z_5^j, z_6^j\}$ represents the set of expected values for the indicator $E_2$ for the $j$th elderly care service personnel to select the elderly. Where $z_1^j$ represents the indicator information of the expectation of the 1st element of the benefit entitlement indicator $E_2$ for the selection of an elder person by the $j$th elderly care service personnel. If the 1st element is required, then $z_1^j = 1$ is taken, otherwise, $z_1^j = 0$. Here, the welfare benefits indicator $E_2$ contains six welfare benefits: "Commuting costs are covered", "Meals are covered", "Separate rooms are available", "Weekly scheduled breaks", "Double pay for holidays" and "Paid vacations".

- $f_3^j = \begin{bmatrix} h_1^j & h_2^j \end{bmatrix}$ represents the matrix of indications for the value of the indicator $E_3$ taken by the $j$th elderly care service personnel when selecting an elder person, $h_r^j \in \{0, 1\}$, $r = 1,2$. Where $h_r^j$ represents the actual instruction information of the $j$th elderly care service personnel for the $r$th element of the work form indicator $E_3$, if the $r$th item is required, then $h_r^j = 1$, otherwise, $h_r^j = 0$. The two job forms included in the work form indicator $E_3$ are: "live-in" and "live-out".

- $f_4^j = \begin{bmatrix} k_1^j & k_2^j & k_3^j & k_4^j & k_5^j & k_6^j \end{bmatrix}$ represents the matrix of indications of the values taken by the $j$th elderly care service personnel for the indicator $E_4$ when selecting an elderly person, where $k_v^j \in \{0, 1\}$, $v = 1,2,\ldots,6$. Where $k_v^j$ represents the desired indicator information for the $v$th item of the elderly situation indicator $E_4$ when the $j$th elderly care service personnel selects an elderly person, and if the $v$th item is needed, then $k_v^j = 1$ is taken, otherwise, $k_v^j = 0$ is taken. Here, the six categories of elderly persons contained in the indicator $E_4$ are: "Self-care male", "Self-care female", "Semi-care male", "Semi-care female", "No care male" and "No care female".

- $G^i = \{g_1^i, g_2^i, g_3^i, g_4^i\}$ represents the set of actual values of each desired indicator of the $i$th elder person for the elderly care service personnel. Among them, $g_1^i$ represents the actual value of for the $i$th elder persons' indicator $E_1$, which is the same as that of $c_2^i$; $g_2^i$ represents the set of actual values for the $i$th elder persons' indicator $E_2$; $g_3^i$ and $g_4^i$ represent the indicator matrix of the actual values of the $i$th elder person for indicator $E_3$, $E_4$.

- $g_2^i = \{z_1^i, z_2^i, z_3^i, z_4^i, z_5^i, z_6^i\}$ represents the actual set of values for the indicator for the $i$th elder person. Where $z_1^i$ represents the indicative information of the 1st element of the indicator $E_2$

of the actual provision of welfare benefits for the $i$th elder person, if the 1st element is actually provided, then $z_1^i = 1$, otherwise, $z_1^i = 0$.

- $g_3^i = \begin{bmatrix} h_1^i & h_2^i \end{bmatrix}$ represents the matrix of indications of the actual values taken by the $i$th elder person for the indicator $E_3$, where $h_r^j \in \{0, 1\}$, $r = 1,2$. Where $h_r^i$ represents the indicator information for the $r$th element of the work form indicator $E_3$ actually provided for the $i$th elder person, and if the $r$th element is actually provided, then $h_r^i = 1$ is taken, otherwise, $h_r^i = 0$, $h_1^i + h_2^i = 1$.

- $g_4^i = \begin{bmatrix} k_1^i & k_2^i & k_3^i & k_4^i & k_5^i & k_6^i \end{bmatrix}$ represents the matrix of actual indications taken for the $i$th elderly person for the elderly situation indicator $E_4$, where $k_v^i \in \{0, 1\}$, $v = 1,2,\ldots,6$,

$\sum_{v=1}^{6} k_v^i = 1$. Where $k_v^i$ represents the actual indication information of the $i$th elder person in response to the $v$th element of the elderly situation indicator $E_4$, if the $v$th content is compliant, then take $k_v^i = 1$, otherwise, $k_v^i = 0$.

- $p_{ij}$ represents the satisfaction of the $i$th elder with the $j$th elderly care service personnel.

- $q_{ij}$ represents the satisfaction of the $j$th elderly care service personnel with the $i$th elderly.

Using the pertinent decision-making data at $A$, $B$, $U$, $E$, $C$, $D$, $F$, and $G$, the problem to be investigated in this paper is how to adopt a workable decision-making approach to solve the problem of matching elderly care service personnel considering multiple types of service expectations.

## 4. Principles and methods

Below is a detailed explanation of the principles and computations of the matching method for elderly care service personnel that takes into account various service expectations.

### 4.1 Description of the expectations of elder people

In the set of indicators $U = \{U_1, U_2, U_3, U_4, U_5, U_6, U_7\}$ that elder people pay attention to when choosing elderly care service personnel, $U_1$ represents age, $U_2$ represents salary, $U_3$ represents work experience, $U_4$ represents education level, $U_5$ represents gender, $U_6$ represents work content, and $U_7$ represents qualification certificate. Here we consider $U_1$, $U_2$, $U_3$, $U_4$ as elastic indicators, the elderly will not directly reject elderly care service personnel when the actual value of the indicator falls short of their expectations; instead, they will be less satisfied with them. These indicators can also be used to gauge how satisfied the elderly are with elderly care service personnel; $U_5$, $U_6$, $U_7$ are inelastic indicators, the elderly will not select the service provider if the elderly care service personnel's actual value falls short of their expectations, and this information can be used to screen out potential replacements. It is important to note that different indicators have different forms of expression. Some indicators can be described by an interval number, such as age ($U_1$). Some indicators can be described by a definite value, such as salary ($U_2$) and work experience ($U_3$). Some indicators can be described by linguistic phrases with hierarchical characteristics, such as education level ($U_4$), for which linguistic phrases with hierarchical characteristics can be transformed into numerical values. Some indicators need to be described by independent language phrases, such as gender ($U_5$) and work content ($U_6$), which can be expressed by an indicator matrix consisting of 0–1 symbols. Some indicators need to be described by independent language phrases together with hierarchical language phrases, such as qualifications ($U_7$), which can be expressed by a numerical matrix.

**4.1.1 Description of the elastic expectations indicator for elder people.** *(1) Age ($U_1$).* According to a survey conducted [56] by a few domestic help companies, elderly people typically have expectations related to their age when selecting elderly care service personnel. For instance, they typically specify a numerical range between 40 and 45 years old. In general, the elderly's expectations on the age index are flexible; that is, when the elderly care service personnel's age and the required age gap between the elderly do not directly reject the service personnel, but rather lower their level of satisfaction; more specifically, when the service personnel's actual age falls within the elderly's expectations of the interval, the elderly's satisfaction with the service personnel is 1. The elder person's satisfaction with the service personnel decreases when the actual age of the personnel exceeds the elder person's expectation taking interval; the elder person's satisfaction with the personnel decreases to zero when the actual age of the personnel is greater than or equal to the maximum limiting age or less than or equal to the minimum limiting age.

Notation $p_{ij}^1$ denotes the satisfaction level of the $i$th elderly people with the age indicator $U_1$ of the $j$th elderly care service personnel, and $d_1^j$ denotes the actual age value of the $j$th elderly care service personnel. According to Feng B and Lai F [57], the formula of $p_{ij}^1$ is shown in Eq (1).

$$p_{ij}^1 = \begin{cases} \dfrac{d_1^j - c_{1\min}}{c_{1\min}^i - c_{1\min}} & c_{1\min} \leq d_1^j < c_{1\min}^i \\ 1 & c_{1\min}^i \leq d_1^j \leq c_{1\max}^i \\ \dfrac{c_{1\max} - d_1^j}{c_{1\max} - c_{1\max}^i} & c_{1\max}^i < d_1^j \leq c_{1\max} \end{cases} \tag{1}$$

Where $c_{1\max}$ denotes the maximum value of the service industry regulations limiting the age of elderly care service personnel, and $c_{1\min}$ denotes the minimum value of the service industry regulations limiting the age of elderly care service personnel. The image of the satisfaction function of elders for the age indicator is shown in Fig 2.

*(2) Wages ($U_2$).* The ability of elderly individuals to pay for elderly care service personnel influences how senior care organizations operate and make managerial decisions. When selecting elderly care service personnel, elder individuals typically focus more on the wage index; however, because different cities have different levels of consumption, different cities have different average salaries for elderly care service personnel. The top five cities in the housekeeping industry salary rankings are Shenzhen, Guangzhou, Shanghai, Wuhan, and Hangzhou, according to 58 Tong Cheng's "China's housekeeping market employment and consumption report" [58]. This indicates that there is a sizable average salary difference between the various cities. In this instance, the elderly provide a specific value to indicate their expectations, and the elderly care service personnel's actual wage requirements are also given a specific value. For instance, the elderly expect 4,500 yuan in compensation from the elderly care service personnel, while the elderly care service personnel demand 5,000 yuan. In general, elderly citizens' expectations regarding the wage index are elastic, meaning that when a service provider's wage demand surpasses their expectations, the elderly will not reject the provider directly, but their level of satisfaction with the provider will decrease. More specifically, this will happen when the provider's actual wage demand value is equal to or less than the elderly citizens' expectations of the wage value, which is 1. The elder person's satisfaction with the elderly care service personnel decreases when the personnel's actual wage demand value exceeds their desired wage value; the elder person's satisfaction with the personnel reaches 0

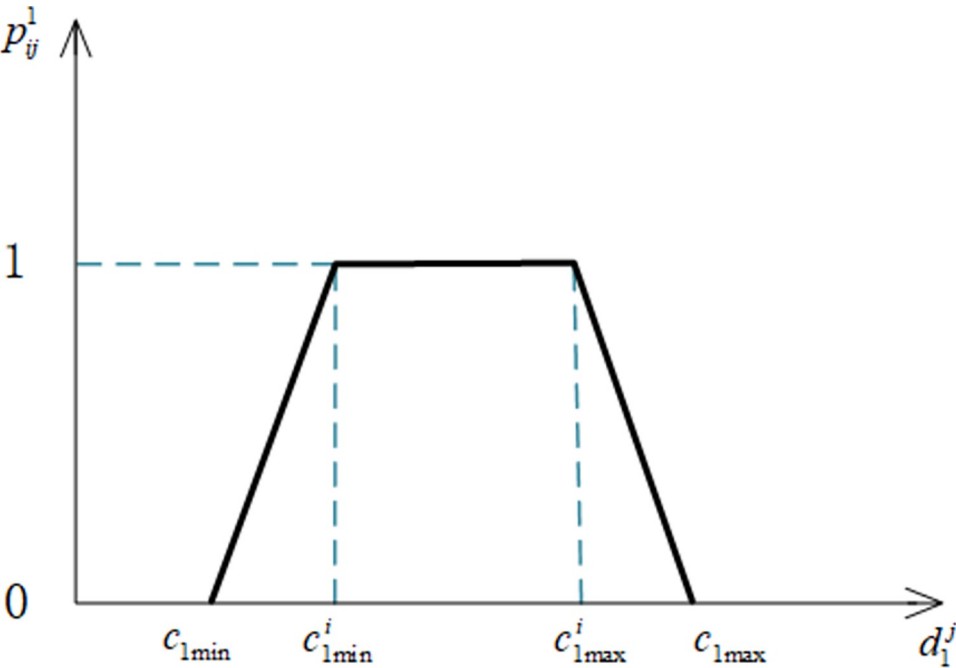

**Fig 2. Satisfaction function of age indicators.**

when the personnel's actual wage demand value exceeds the maximum value of the wage that the elder person can accept.

Notation $p_{ij}^2$ denote the satisfaction level of the $i$th elderly with the wage indicator $U_2$ of the $j$th elderly care service personnel, where $c_2^i$ denotes the expectation of the $i$th elderly for the wage indicator of the elderly care service personnel, and $d_2^j$ denotes the actual value of the $j$th elderly care service personnel for the wage indicator. The formula of $p_{ij}^2$ is shown in Eq (2).

$$p_{ij}^2 = \begin{cases} 1 & d_2^j \le c_2^i \\ \dfrac{c_{2\max}^i - d_2^j}{c_{2\max}^i - c_2^i} & c_2^i < d_2^j < c_{2\max}^i \\ 0 & d_2^j \ge c_{2\max}^i \end{cases} \tag{2}$$

Where $c_{2\max}^i$ denotes the maximum value of the wage indicator value acceptable to the $i$th elder. The image of the satisfaction function of the elderly for the wage indicator is shown in Fig 3.

*(3) Work experience ($U_3$).* When selecting an elderly care service personnel, elder adults typically have a preference for those with more work experience. Here, we take into account gauging the length of time that elderly care service personnel have worked in the field in order to assess their work experience [59]. Elderly people's expectations of the work experience indicator are generally elastic, meaning that when the elderly care service personnel's work experience falls short of their expectations, they will not reject the staff directly, but rather will be less satisfied; in particular, when the staff's actual work experience matches or exceeds the elder people's expectations of the work experience indicator years, the elder people's satisfaction with the staff is 1. The elder people's satisfaction with the elderly care service personnel will decline when the actual working years of the personnel are less than the expected years of the

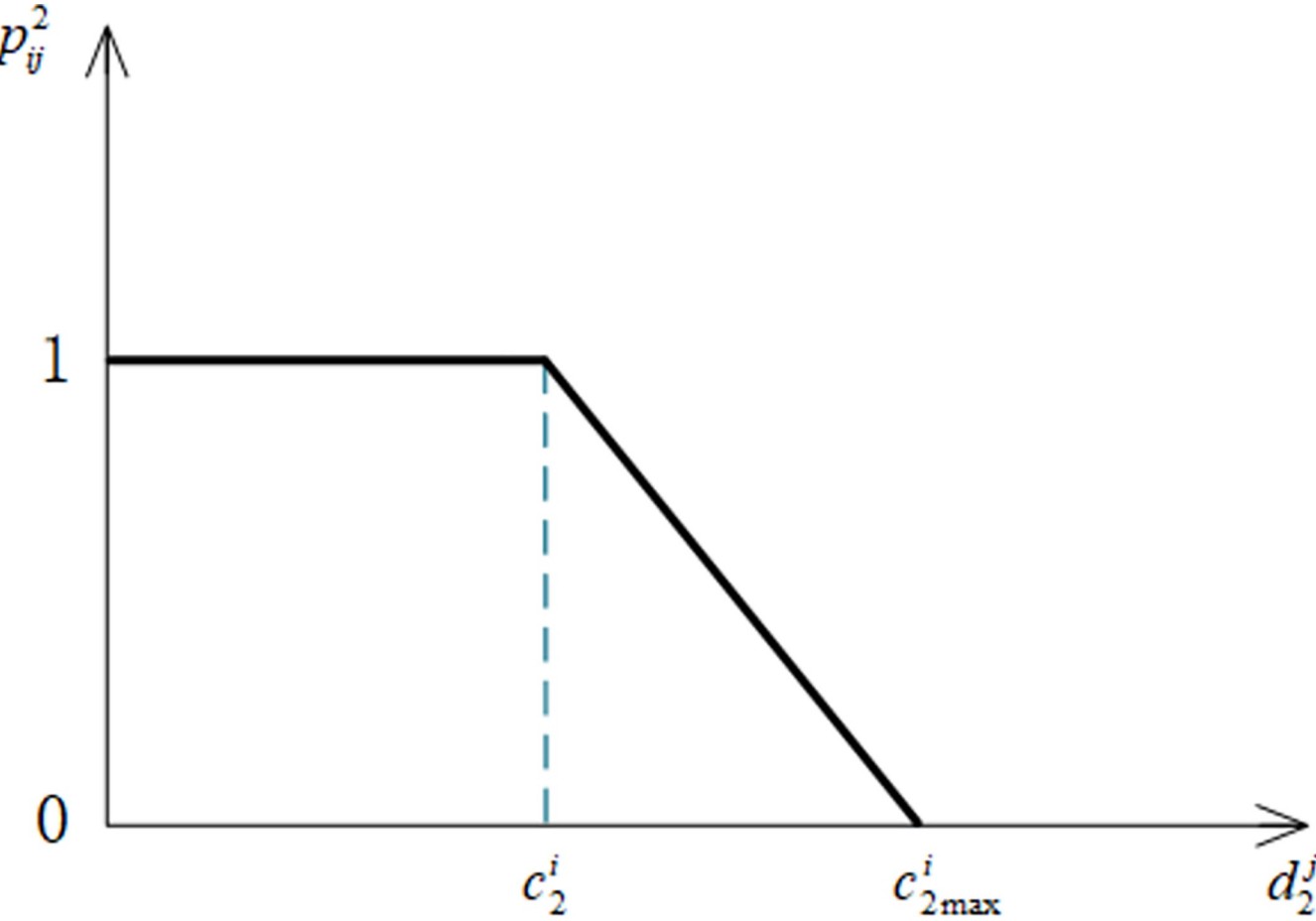

**Fig 3. Satisfaction function of wage indicators.**

elder people on the work experience indicator; the elder people's satisfaction with the service personnel will drop to zero when the actual working years of the personnel are 0.

Notation $p_{ij}^3$ denotes the satisfaction of the $i$th elder person with the work experience indicator $U_3$ of the $i$th elderly care service personnel, where $c_3^i$ denotes the expectation of the $i$th elder person with respect to the work experience indicator of the elderly care service personnel, and $d_3^j$ denotes the actual value of the $j$th elderly care service personnel with respect to the work experience indicator. The formula of $p_{ij}^3$ for calculating this indicator is shown in Eq (3).

$$p_{ij}^3 = \begin{cases} \dfrac{d_3^j}{c_3^i} & d_3^j < c_3^i \\ 1 & d_3^j \geq c_3^i \end{cases} \tag{3}$$

The image of the work experience indicator satisfaction function is shown in Fig 4.

*(4) Educational level ($U_4$).* When selecting elderly care service personnel, elders typically consider the education level of those employees as well, and they typically select those with higher educational backgrounds [60]. In general, the elderly's expectations regarding the education level indicator are elastic; that is, if the service personnel's actual education level falls short of the elderly's expectations, the elderly will not outright reject the service personnel but will instead express less satisfaction with the personnel. In particular, the elderly's satisfaction

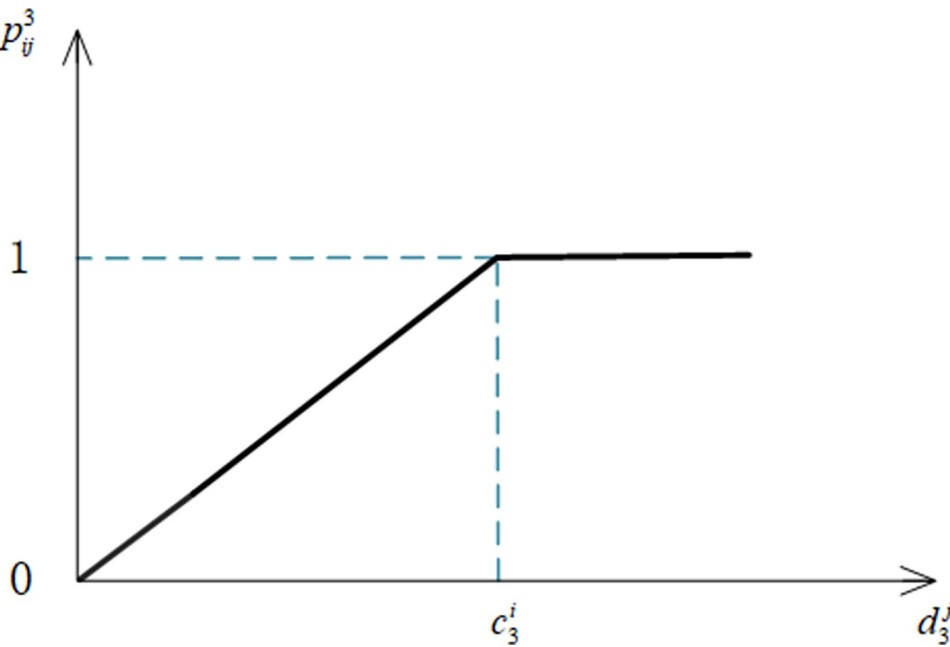

**Fig 4. Satisfaction function of work experience indicators.**

with the service personnel is 1 when the actual education level of the personnel is higher than or equal to the elderly's expectation value for the education level indicator. The elderly's satisfaction with the elderly care service personnel will decline when the actual education level of the personnel is lower than the elderly's expectation value for the education level indicator; the elderly's satisfaction with the service personnel will drop to 0 when the education level of the personnel is 0.

Considering that the value of the education level indicator has a hierarchical character, according to Zuo Meiyun [61], the education level is divided into levels, and the value range of the education level indicator is recorded as $c_4^i, d_4^j \in \{0, 1, 2, 3, 4, 5\}$, in which 0 stands for illiteracy, 1 stands for elementary school, 2 stands for middle school, 3 stands for high school/secondary school, 4 stands for specialist/higher vocational school, and 5 stands for bachelor's degree and above. $p_{ij}^4$ denotes the satisfaction of the $i$th elder person with the $j$th elderly care service personnel's education level indicator $U_4$, where $c_4^i$ indicates the expectation value of the $i$th elder person for the service personnel education level indicator, and $d_4^j$ indicates the actual value of the $j$th elderly care service personnel for the education level indicator. The formula of $p_{ij}^4$ for calculating this indicator is shown in Eq (4).

$$p_{ij}^4 = \begin{cases} \dfrac{d_4^j}{c_4^i} & d_4^j < c_4^i \\ 1 & d_4^j \geq c_4^i \end{cases} \tag{4}$$

An indicative graph of the satisfaction function image for the educational attainment indicator is shown in Fig 5.

**4.1.2 Description of the indicator of inelastic expectations of elders.** *(1) Gender ($U_5$).* Depending on their personal circumstances, elders will specify the gender of the elderly care service personnel when selecting them. According to online and offline survey statistics [61], elder people have inelastic expectations about gender indicators. This means that if the

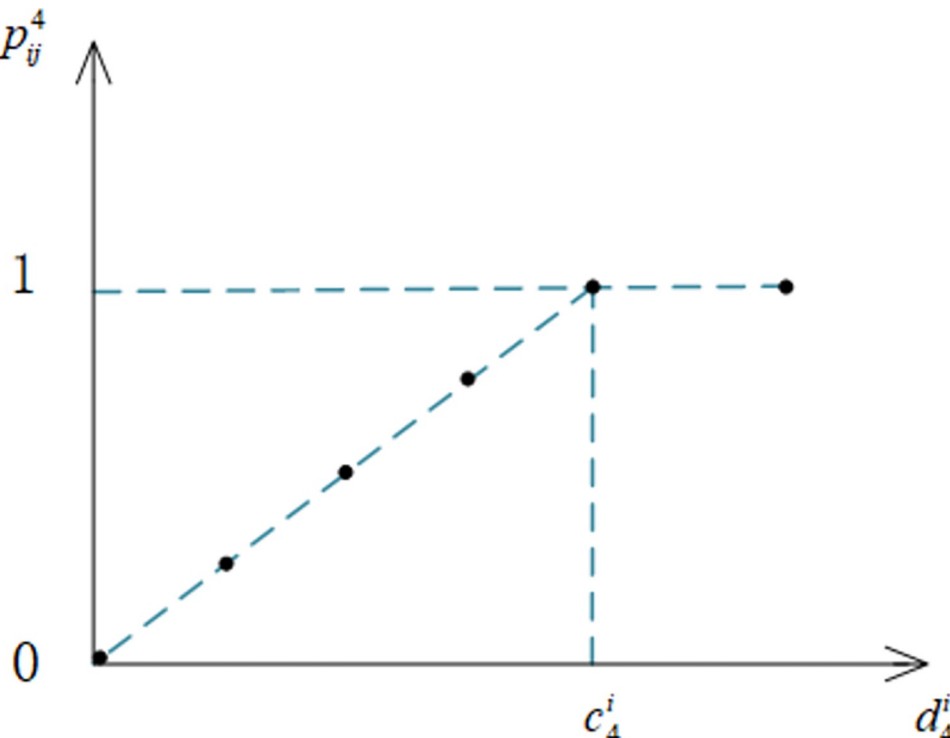

**Fig 5. Indicative graph of satisfaction function for the indicator of educational attainment.**

expected and actual genders of the service personnel differ, the elder people will refuse to choose the service personnel. As a result, this indicator can be used as a constraint to screen the service personnel. The expected gender of the elderly person and the actual gender of the service personnel are represented in this instance by the 0–1 value indicator matrix.

Recall that $c_5^i = \begin{bmatrix} sex_1^i & sex_2^i \end{bmatrix}$ denotes the matrix of desired values for the indicator $U_5$ for the selection of elderly care service personnel for the $i$th elder person, $d_5^j = \begin{bmatrix} sex_1^j & sex_2^j \end{bmatrix}$ denotes the matrix of actual values for the indicator $U_5$ for the $j$th elderly care service personnel, $sex_y^i, sex_y^j \in \{0, 1\}, y = 1,2$. The value of $sex_y^i - sex_y^j$ is used to determine whether the actual gender of the $j$th elderly care service personnel meets the desired gender needs of the $i$th elder adult. When there are $sex_y^i - sex_y^j < 0$ and $y = 1,2$, it means that the actual gender of the $j$th elderly care service personnel does not meet the expected gender needs of the $i$th elderly, the $i$th elderly will not choose this service personnel. When there are $sex_y^i - sex_y^j \geq 0$ and $y = 1,2$, it means that the actual gender of the $j$th elderly care service personnel meets the expected gender needs of the $i$th elderly, the $j$th elderly care service personnel can be used as an alternative to the $i$th elderly with respect to the gender indicator.

*(2) Job description ($U_6$).* Based on their individual circumstances, the elderly will specify exactly what they need from the elderly care service personnel when selecting one. In general, the elderly have inelastic expectations for the work content indicator. This means that if the service personnel's actual work content falls short of their expectations, the elderly will not choose them. As a result, this indicator can be used as a constraint to screen the service personnel. This metric can be employed as a limitation when vetting employees. The job content that the elderly expect from the service personnel and the actual job content that they can provide are represented in this instance by a matrix of 0–1 values.

Notation $c_6^i = \begin{bmatrix} l_1^i & l_2^i & l_3^i & l_4^i & l_5^i & l_6^i & l_7^i \end{bmatrix}$ denotes the indicator matrix of the value of the content indicator of the elderly care service personnel for the $i$th elder person, $d_6^j = \begin{bmatrix} l_1^j & l_2^j & l_3^j & l_4^j & l_5^j & l_6^j & l_7^j \end{bmatrix}$ denotes the indicator matrix of the value of the actual content indicator of the $j$th service personnel, $l_s^i, l_s^j \in \{0,1\}$, $s = 1,2,\ldots,7$. The value of $l_s^j - l_s^i$ is used to determine whether the work content actually available to the $j$th service worker meets the work content expected by the $i$th elder. When there are $l_s^j - l_s^i < 0$ and $s = 1,2,\ldots,7$, it means that the actual work content provided by the $j$th service personnel cannot meet the work content expected by the $i$th elderly, the $i$th elderly will not choose this service personnel; when there are $l_s^j - l_s^i \geq 0$ and $s = 1,2,\ldots,7$, it means that the actual work content provided by the $j$th service personnel can meet the work content demand expected by the $i$th elderly, for the indicator of work content the $j$th service personnel can be an alternative for the $i$th elderly.

*(3) Certificates of qualification ($U_7$).* The elderly will insist on the credentials possessed by the elderly care service personnel when selecting one. In general, the elderly's expectations regarding the qualification index are an inelastic indicator. This means that if the service personnel's actual qualification certificate and grade do not meet the elderly's expectations, the elderly will refuse to choose the service personnel. As a result, this indicator can be used as a constraint to screen the service personnel. The value indication matrix in this case represents the values of the qualification indexes that the elderly expected and the actual qualification indexes of the service personnel.

The relevant documents and bibliographies at [62, 63] contain the specific occupational functions and job content of the various types of qualifications that the service personnel can hold. The necessary qualifications and grades are provided by inquiring about the particular needs of the elderly, as some of them are incapable of making specific requests for their credentials or grades.

Note that $c_7^i = \begin{bmatrix} o_1^i & o_2^i & o_3^i & o_4^i & o_5^i & o_6^i \end{bmatrix}$ represents the matrix of indications of the values of the credentials indicator for the $i$th elder-to-service-personnel indicator, and $d_7^j = \begin{bmatrix} o_1^j & o_2^j & o_3^j & o_4^j & o_5^j & o_6^j \end{bmatrix}$ represents the matrix of indications of the values of the actual credentials indicator for the $j$th service personnel. The six credentials included in the credentials indicator $U_7$ are: "Domestic Helper Certificate", "Household Service Worker Certificate", "Home Care Worker Certificate", "Elderly Care Worker Certificate", "Dietitian Certificate" and "Chef Certificate". Here the values of $o_t^i$ and $o_t^j$ indicate the level of the qualification index $U_7$, $o_t^i, o_t^j \in \{0, 1, 2, 3, 4, 5\}$ and $t = 1,2,\ldots,6$, where the highest level of the "Household Service Worker Certificate" and "Home Care Worker Certificate" is level 5; the highest level of the "Elderly Care Worker Certificate" and "Dietitian Certificate" is level 4; the highest level of the "Domestic Helper Certificate" is level 3; and the highest level of the "Chef Certificate" is level 1.

The value of $o_t^j - o_t^i$ is used to determine whether the qualification certificate and level actually held by the $j$th elderly care service worker meets the qualification certificate and level expected by the $i$th elderly, $t = 1,2,\ldots,6$. When there are $o_t^j - o_t^i < 0$ and $t = 1,2,\ldots,6$, it means that the qualification certificate and grade of the $j$th service personnel does not meet the qualification certificate and grade expected by the $i$th elderly, that is, the $i$th elderly will not choose this service personnel; when there are $o_t^j - o_t^i \geq 0$ and $t = 1,2,\ldots,6$, it means that the qualification certificate and grade of the $j$th service personnel meets the qualification certificate and grade expected by the $i$th elderly, that is, for the qualification certificate indicator, the $j$th service personnel can be the alternative for the $i$th elderly. That is to say, the service worker can be an alternative for the elder person in terms of the qualification indicator.

**4.1.3 Calculation of overall satisfaction of elders.** Each indicator has a different weight assigned by different elderly people, and the weights are used to show how much weight elder

people assign to each indicator. Here $w_i^1$, $w_i^2$, $w_i^3$, $w_i^4$ are used to indicate the degree of importance that the $i$th elder person attaches to the indicators $U_1$, $U_2$, $U_3$, $U_4$ and the values are given by each elder person.

Noting that $p_{ij}$ denotes the overall satisfaction of $i$th elder adults with the $j$th elderly care service providers, $p_{ij}$ is calculated using the Eq (5).

$$p_{ij} = w_i^1 p_{ij}^1 + w_i^2 p_{ij}^2 + w_i^3 p_{ij}^3 + w_i^4 p_{ij}^4 \tag{5}$$

which is $w_i^1 + w_i^2 + w_i^3 + w_i^4 = 1$.

## 4.2 Description of expectations of elderly care service personnel

The set of indicators that elderly care service personnels consider to be taken into account when selecting an elder is $E = \{E_1, E_2, E_3, E_4\}$. Among them, $E_1$ stands for wages, $E_2$ for welfare benefits, $E_3$ for work forms, and $E_4$ for the situation of the elderly. Here we consider $E_1$ and $E_2$ as elastic indicators, these indicators can be used to gauge how satisfied service personnels are with alternative elderly populations when the actual value of the indicator falls short of expectations. In this scenario, the staff members will not decline to serve the elderly directly, but rather will express less satisfaction with them. $E_3$ and $E_4$ are inelastic indicators, such indicators can be utilized for evaluating alternative elder, if the actual value of the indicator does not meet the desired requirements, the service personnel will decline to choose to serve the elder. It is crucial to remember that various indicators have distinct ways of being expressed and that some indicators have defined values that can be used to characterize them, such as wages ($E_1$); some indicators need to be described by independent language phrases, such as welfare benefits ($E_2$), which can be represented by a set of 0–1 symbols; some indicators need to be described by independent language phrases, such as work forms ($E_3$) and the situation of the elderly ($E_4$), which can be represented by a set of 0–1 symbols.

**4.2.1 Description of the elastic expectations indicator for elder care service personnel.** *(1) Wages ($E_1$).* When choosing an elder, senior care providers will undoubtedly take their wage level into account. In general, the expectations of the service personnel regarding the wage indicator are flexible. This means that in situations where the wage value provided by the elderly falls short of the service personnel's expected wage value, the service personnel will not decline to serve the elderly directly, but rather will lessen their satisfaction. To be more precise, in situations where the wage value provided by the elderly meets or exceeds the service personnel's expected wage value, the service personnel's satisfaction with the elderly will be 1; in situations where the wage value provided by the elderly falls short of the service personnel's desired wage value, the service personnel's satisfaction with the elderly. The service personnel's satisfaction with the elderly person drops to zero when the older person's offered wage value is equal to or less than the minimum value of the service person's acceptable wage.

Let $q_{ij}^1$ denote the satisfaction of the $j$th service provider with the wage indicator $E_1$ for the $i$th elder, where $g_1^i$ denotes the actual value of the wage indicator for the $i$th elder, and $f_1^j$ denotes the expected value of the wage demand for the $j$th service personnel. The formula of $q_{ij}^1$ is shown in Eq (6).

$$q_{ij}^1 = \begin{cases} 0 & g_1^i \leq f_{1\min}^j \\ \dfrac{g_1^i - f_{1\min}^j}{f_1^j - f_{1\min}^j} & f_{1\min}^j < g_1^i < f_1^j \\ 1 & g_1^i \geq f_1^j \end{cases} \tag{6}$$

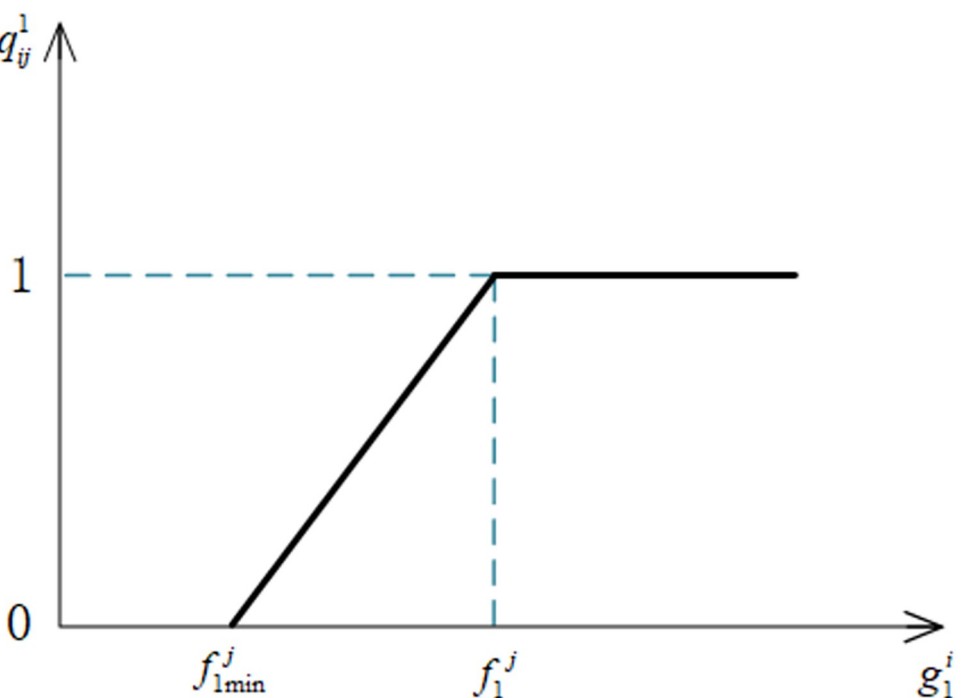

**Fig 6. Satisfaction function of wage indicators for service personnel.**

Where $f^j_{1\min}$ denotes the minimum value of the wage indicator that is acceptable to the $j$th service personnel. The image of the satisfaction function of the wage indicator for service personnel is shown in Fig 6.

*(2) Welfare benefits ($E_2$).* When choosing an elder, elderly care service personnels also take into account the advantages that an elder individual can provide. In general, service personnel's expectations of the welfare treatment indicator are flexible; that is, when the elderly's welfare treatment falls short of what the staff expects, the staff will not refuse to serve the elderly directly, but rather will decrease their level of satisfaction. More specifically, the elderly's welfare treatment may be reflected in a variety of specific ways, depending on what the staff expects to be fully provided for the elderly's welfare treatment. The level of satisfaction that service personnel have with the elderly is 1 when the elderly can provide all the welfare benefits that the personnel have expected; it will decrease when the elderly can only provide a portion of the welfare benefits that the personnel have expected; and it will drop to 0 when the elderly are unable to provide the welfare benefits that the personnel have expected. Here, aggregation is used to show indicators of the welfare benefits that elder actually provide versus those that service personnels desire.

Notation $q^2_{ij}$ Indicates the $j$th service personnel's satisfaction with the $i$th elder's welfare entitlement indicator $E_2$, and $q^2_{ij}$ is calculated using the Eq (7).

$$q^2_{ij} = \frac{f^j_2 \cap g^i_2}{f^j_2} \tag{7}$$

**4.2.2 Description of indicators of inelastic expectations of service personnel.** *(1) Form of work ($E_3$).* When choosing an older individual, elderly service providers will need to know their employment history. In general, the service staff's expectations regarding the work form

indicator are inelastic, meaning that if the elder's work form does not match the service staff's expectations, the latter will decline to assist the former. As a result, this indicator can be used as a screening tool for older individuals. The work form required by the elderly and the job form anticipated by the support staff are represented in this instance by the 0–1 value indicator matrix.

Recall that $f_3^j = \begin{bmatrix} h_1^j & h_2^j \end{bmatrix}$ represents the matrix of indications of the values of the indicator $E_3$ for the selection of the elder by the $j$th service provider, $g_3^i = \begin{bmatrix} h_1^i & h_2^i \end{bmatrix}$ represents the matrix of indications of the actual values of the indicator $E_3$ for the $i$th elder person, $h_r^j, h_r^i \in \{0, 1\}$, $r = 1,2$, $h_1^i + h_2^i = 1$. The value of $h_r^j - h_r^i$ is used to determine whether the form of work required by the $i$th elder corresponds to the form of work expected by the $j$th service provider. When there are $h_r^j - h_r^i < 0$, $r = 1,2$, it means that the form of work needed by the $i$th elder does not correspond to the form of work expected by the $j$th service personnel, the $j$th service personnel will not choose to serve this elder. When $h_r^j - h_r^i \geq 0$, $r = 1,2$, it means that the form of work needed by the $i$th elder corresponds to the form of work expected by the $j$th service personnel, for the indicator of the form of work the $i$th elder can be an alternative to the $j$th service provider.

*(2) Situation of the elderly ($E_4$).* When choosing elder clients, service personnels will base their decisions on certain conditions. In general, the expectations of service personnel regarding the elderly situation indicator are inelastic, meaning that they will decline to serve the elderly if their actual situation does not align with their expectations. This means that the elderly situation indicator can be used as a screening tool. Here, a 0–1 value indicator matrix represents the service staff's perception of the senior population and their actual circumstances.

Notation $f_4^j = \begin{bmatrix} k_1^j & k_2^j & k_3^j & k_4^j & k_5^j & k_6^j \end{bmatrix}$ denotes the matrix of indications of the values of the indicator $E_4$ for the selection of the elderly by the service personnel of the $j$th service, $g_4^i = \begin{bmatrix} k_1^i & k_2^i & k_3^i & k_4^i & k_5^i & k_6^i \end{bmatrix}$ denotes the matrix of indications of the actual values of the indicator $E_4$ for the situation of the elderly of the $i$th service, $k_v^j, k_v^i \in \{0, 1\}$, $v = 1,2,\ldots,6$, $\sum_{v=1}^{6} k_v^i = 1$.

The value of $k_v^j - k_v^i$ is used to determine whether the actual situation of the $i$th elderly matches that the $j$th service personnel expects to serve. When there are $k_v^j - k_v^i < 0$, $v = 1,2,\ldots,6$, it means that the actual situation of the $i$th elderly does not match with the situation of the elderly that the $j$th service provider expects to serve, the $j$th service provider will not choose to serve this elderly. When $k_v^j - k_v^i \geq 0$, $v = 1,2,\ldots,6$, it means that the actual situation of the $i$th elderly matches with the situation of the elderly that the $j$th service provider expects to serve, for the indicators of the situation of the elderly, the $i$th elderly can be used as an alternative for the $j$th service provider.

**4.2.3 Calculation of overall satisfaction of service personnel.**   Each indicator has a distinct level of relevance assigned by different service workers, and the weights are used to show how much weight each indicator has. Here, $w_j^1$ and $w_j^2$ are used to indicate the degree of importance given by the $j$th service personnel to the indicators $E_1$ and $E_2$, and the values are given by each service personnel.

Noting that $q_{ij}$ denotes the overall satisfaction of the $j$th service provider with the $i$th elder adult, $q_{ij}$ is calculated using the formula:

$$q_{ij} = w_j^1 q_{ij}^1 + w_j^2 q_{ij}^2 \tag{8}$$

which is $w_j^1 + w_j^2 = 1$.

## 4.3 Matching elders with service providers

The matching model is constructed in a targeted manner for each of the two typical scenarios that may occur in reality, namely, sufficient or insufficient service personnel, and the model is solved to yield the best matching solution.

(1) The bilateral matching optimization model is built with the intention of maximizing both senior and service personnel satisfaction when the number of seniors is less than or equal to the number of service personnel. Should the model have a solution, matching will be successful, and each senior will have a suitable service personnel match; if not, the seniors will be asked if they would like to lower the requirements of the inelastic indicators, and if any of them do so, they will be re-matched with the service personnel. In the event that a solution cannot be found, the elderly are asked if they would like to reduce their requirements for the inelastic indicators. Should they agree to do so, they will be paired with service personnel once more; if not, the computation moves on to the second phase, which takes the elderly priority model into account.

(2) If there are more elderly people than there are service personnel, the elderly people's basic indexes are used to determine priority. After that, the bilateral matching optimization model is constructed with the aim of maximizing satisfaction between the elderly people and the service personnel, taking into account the elderly people's priority. The elderly people who are not successful are then asked if they would like to lower the requirements of the inelastic indexes of the service personnel. If some of them do so, they will be matched with the service personnel again. Should an elder opt to lower their criteria, they will be paired with service people once more. If no elder chooses to lower their inelastic index requirements, the matching process will come to an end, and the successful pairing of a select few seniors will be achieved.

The matching process is shown in Fig 7.

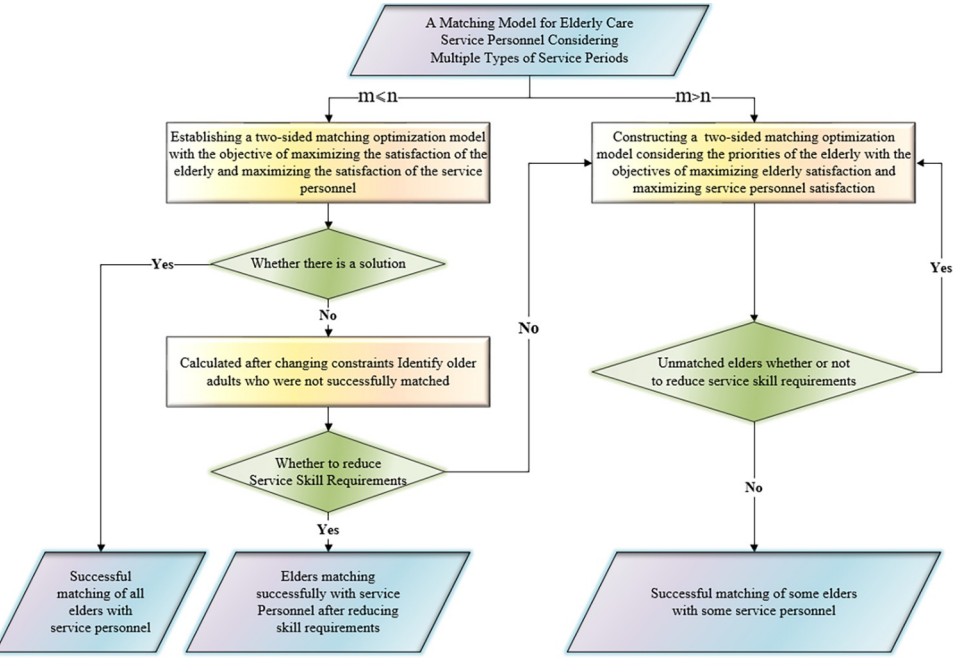

**Fig 7. Matching flowchart.**

**4.3.1 Construction and solution of the matching model for the case of sufficient service personnel ($m \leq n$).** Let $x_{ij}$ be a 0–1 type decision variable, $x_{ij} = 1$ means that the elderly $A_i$ and the service personnel $B_j$ form a match; otherwise $x_{ij} = 0$. Based on the satisfaction $p_{ij}$ of the elderly to the service personnel and the satisfaction $q_{ij}$ of the service personnel to the elderly, we can construct the following bilateral matching optimization model with the objective of maximizing the satisfaction of the elderly and the satisfaction of the service personnel:

$$\max Z_1 = \sum_{i=1}^{m} \sum_{j=1}^{n} p_{ij} x_{ij} \tag{9A}$$

$$\max Z_2 = \sum_{i=1}^{m} \sum_{j=1}^{n} q_{ij} x_{ij} \tag{9B}$$

$$s.t. \ (sex_y^i - sex_y^j)x_{ij} \geq 0, y = 1, 2 \tag{9C}$$

$$(l_s^j - l_s^i)x_{ij} \geq 0, s = 1, 2, \ldots, 7 \tag{9D}$$

$$(o_t^j - o_t^i)x_{ij} \geq 0, t = 1, 2, \ldots, 6 \tag{9E}$$

$$(h_r^j - h_r^i)x_{ij} \geq 0, r = 1, 2 \tag{9F}$$

$$(k_v^j - k_v^i)x_{ij} \geq 0, v = 1, 2, \ldots, 6 \tag{9G}$$

$$\sum_{j=1}^{n} x_{ij} = 1 \tag{9H}$$

$$\sum_{i=1}^{m} x_{ij} \leq 1 \tag{9I}$$

$$i = 1, 2, \ldots, m; j = 1, 2, \ldots, n \tag{9J}$$

In model (9), Eq (9A) and Eq (9B) are the objective functions, and Eq (9A) indicates that the elder's satisfaction with the service personnel is maximized; Eq (9B) indicates that the service personnel's satisfaction with the elder is maximized. Constraint (9c) indicates that the gender of the $j$th service personnel must meet the gender needs of the $i$th elderly; constraint (9d) indicates that the work content of the $j$th service personnel must meet the work content of the $i$th elderly; constraint (9e) indicates that the qualification certificates and grades of the $j$th service personnel must meet the needs of the $i$ th elderly; constraint (9f) indicates that the work form of the $j$th service personnel must meet the needs of the $i$th elderly; constraint (9g) means that the actual situation of the $i$th elder is within the situation of the elder person served by the $j$th service personnel; constraint (9h) means that each elder must be matched with one service personnel only and only one service personnel; and Constraint (9i) means that each service worker can only be matched with one elder person or not matched with an elder person.

Model (9) is a bi-objective 0–1 integer programming model that can be solved using a linear weighting method [64], the basic idea of which is to transform a multi-objective problem into

an algorithm for single-objective planning by summing the objective functions in a linearly weighted manner. Model (9) is transformed into the following single-objective optimization model:

$$\max Z = \omega_1 \sum_{i=1}^{m} \sum_{j=1}^{n} p_{ij} x_{ij} + \omega_2 \sum_{i=1}^{m} \sum_{j=1}^{n} q_{ij} x_{ij} \tag{10A}$$

$$s.t. \quad (sex_y^i - sex_y^j) x_{ij} \geq 0, y = 1, 2 \tag{10B}$$

$$(l_s^j - l_s^i) x_{ij} \geq 0, s = 1, 2, \ldots, 7 \tag{10C}$$

$$(o_t^j - o_t^i) x_{ij} \geq 0, t = 1, 2, \ldots, 6 \tag{10D}$$

$$(h_r^j - h_r^i) x_{ij} \geq 0, r = 1, 2 \tag{10E}$$

$$(k_v^j - k_v^i) x_{ij} \geq 0, v = 1, 2, \ldots, 6 \tag{10F}$$

$$\sum_{j=1}^{n} x_{ij} = 1 \tag{10G}$$

$$\sum_{i=1}^{m} x_{ij} \leq 1 \tag{10H}$$

$$i = 1, 2, \ldots, m; j = 1, 2, \ldots, n \tag{10I}$$

The weights $\omega_1$ and $\omega_2$ reflect the importance of both subjects in the actual matching decision. If $\omega_1 > \omega_2$, it means that the satisfaction degree of the subject on the elderly side should be emphasized in the matching decision; if $\omega_1 < \omega_2$, it means that the satisfaction degree of the subject on the service personnel side should be emphasized in the matching decision; and if $\omega_1 = \omega_2 = 0.5$, it means that the fairness of both subjects should be considered in the matching decision.

The model (9) can be solved using optimization tools such as LINGO 18.0, Cplex, Matlab, and others after being converted into a single-objective optimization model (10) via the linear weighting approach. The fact that this model might not have a solution should be noted. If it does, there could be two possible causes: (1) the elderly have very high expectations in terms of their inelastic index, which makes it impossible for service personnel to meet; or (2) there are multiple elderly individuals with inelastic index requirements that can only be satisfied by a smaller number of service personnel, but a service personnel can only assist one elderly person at a time. In the event that this model cannot be solved, modify the constraint that requires a service personnel to be matched for every elder in the original model. Change constraint (9h) to $\sum_{j=1}^{n} x_{ij} \leq 1$. Resolve the model to determine which particular elder has not been matched successfully. Ask the elder who has not been matched successfully if they would like to lessen the requirement of the service personnel's inelastic indicator, which is the qualification certificate ($U_7$) indicator requirement. Proceed to Section 4.3.2 for calculation if the elderly agree to

reduce the requirements. If the original optimization model cannot be solved, repeat the process until an answer is found.

**4.3.2 Construction and solution of matching model for service personnel insufficiency scenario ($m>n$).** The elderly's priority is taken into account when A>B or the second case cannot be resolved. This is primarily reflected in the three areas of the elderly's age, ability to care for themselves, and living situation. Elderly people who are older have a higher priority than those who are younger; elderly people who do not take care of themselves have a higher priority than those who are semi-autonomous and higher than those who do; elderly people who live alone have a higher priority than those who live with their spouses and above those who live with their children. The formula used to determine the elderly's priority is provided below.

The formula for calculating the age priority of seniors is shown in Eq (11).

$$agc_i = \frac{ag_i - ag_{\min}}{ag_{\max} - ag_{\min}} \tag{11}$$

Where $agc_i$ denotes the age-priority score of the $i$th elder person and $agc_i \in [0,1]$; $ag_{\min} = \min\{ag_i | i = 1, 2, \ldots, m\}$ denotes the minimum age among the $i$th elder; and $ag_{\max} = \max\{ag_i | i = 1, 2, \ldots, m\}$ denotes the maximum age among the $i$th elder.

The formula for calculating the priority of self-care for the elderly is shown in Eq (12).

$$scc_i = \frac{sc_i - sc_{\min}}{sc_{\max} - sc_{\min}} \tag{12}$$

Among them, $scc_i$ represents the self-care ability priority score of the $i$th elderly and $scc_i \in [0,1]$; $sc_{\min} = \min\{sc_i \mid i = 1, 2, \ldots, m\}$ represents the minimum value of self-care ability score among $i$ elderly; $sc_{\max} = \max\{sc_i \mid i = 1, 2, \ldots, m\}$ represents the maximum value of self-care ability score among $i$ elderly. Here it is assumed that when the $i$th elder's self-care ability is not self-care ability, the $i$th elder's self-care ability score is $sc_i = 3$; when the $i$th elder's self-care ability is semi-self-care ability, the $i$th elder's self-care ability score is $sc_i = 2$; and when the self-care ability of the $i$th elder is self-care ability, the score of the $i$th elder's self-care ability is $sc_i = 1$.

The formula for calculating the priority of elderly citizens' living situations is shown in Eq (13).

$$lcc_i = \frac{lc_i - lc_{\min}}{lc_{\max} - lc_{\min}} \tag{13}$$

Among them, $lcc_i$ represents the living situation priority score of the $i$th elderly and $lcc_i \in [0,1]$; $lc_{\min} = \min\{lc_i \mid i = 1, 2, \ldots, m\}$ represents the minimum value of living situation priority score among $i$ elderly; $lc_{\max} = \max\{lc_i \mid i = 1, 2, \ldots, m\}$ represents the maximum value of living situation priority score among $i$ elderly. Here it is set that when the living situation of the $i$th elderly is "living alone and not living in the same city as his children", the score of the $i$th elder's living situation is $lc_i = 5$; when the living situation of the $i$th elderly is "living alone and in the same city with their children", the score of the $i$th elderly's living situation is $lc_i = 4$;When the $i$th elderly's living situation is "living with his spouse and not in the same city as his children", the score of the $i$th elderly's living situation is $lc_i = 3$; When the living situation of the $i$th elderly is "living with the spouse and the same city with the children", the score of the $i$th elderly's residence status is $lc_i = 2$; When the $i$th elderly's living situation is "living with children", the $i$th elderly's living situation score is $lc_i = 1$.

The formula for calculating the priority $hcc_i$ of the physical condition of the elderly is shown in Eq (14).

$$pr_i = \frac{agc_i + scc_i + lcc_i}{3} \tag{14}$$

Let $x_{ij}$ be a 0–1 decision variable, and $x_{ij} = 1$ means that the elderly $A_i$ and the service personnel $B_j$ form a match; otherwise $x_{ij} = 0$. Based on the satisfaction of the elderly to the service personnel $p_{ij}$ and the satisfaction $q_{ij}$ of the service personnel to the elderly, as well as the priority $pr_i$ of the elderly. With the aim of optimizing both the service personnel's and the elderly's satisfaction, while considering the elderly's priority, we build the bilateral matching optimization model is shown in Model (15).

$$\max Z_1 = \sum_{i=1}^{m} \sum_{j=1}^{n} pr_i p_{ij} x_{ij} \tag{15A}$$

$$\max Z_2 = \sum_{i=1}^{m} \sum_{j=1}^{n} q_{ij} x_{ij} \tag{15B}$$

$$s.t. \ (sex_y^i - sex_y^j)x_{ij} \geq 0, y = 1, 2 \tag{15C}$$

$$(l_s^j - l_s^i)x_{ij} \geq 0, s = 1, 2, \ldots, 7 \tag{15D}$$

$$(o_t^j - o_t^i)x_{ij} \geq 0, t = 1, 2, \ldots, 6 \tag{15E}$$

$$(h_r^j - h_r^i)x_{ij} \geq 0, r = 1, 2 \tag{15F}$$

$$(k_v^j - k_v^i)x_{ij} \geq 0, v = 1, 2, \ldots, 6 \tag{15G}$$

$$\sum_{j=1}^{n} x_{ij} \leq 1 \tag{15H}$$

$$\sum_{i=1}^{m} x_{ij} \leq 1 \tag{15I}$$

$$i = 1, 2, \ldots, m; j = 1, 2, \ldots, n \tag{15J}$$

In model (15), Eq (15A) and Eq (15B) are the objective functions, and Eq (15A) indicates that elders are most satisfied with service personnel after considering prioritization; Eq (15B) indicates that the service personnel's satisfaction with the elder is maximized. Constraint (15c) indicates that the gender of the $j$th service personnel must meet the gender needs of the $i$th elderly; constraint (15d) indicates that the work content of the $j$th service personnel must meet the work content of the $i$th elderly; constraint (15e) indicates that the qualification certificates and grades of the $j$th service personnel must meet the needs of the $i$th elderly; constraint (15f) indicates that the work form of the $j$th service personnel must meet the needs of the $i$th elderly; constraint (15g) means that the actual situation of the $i$th elder is within the situation of the elder person served by the $j$th service personnel; constraints (15h) indicate that each elder is

matched with only one personnel or no personnel; and Constraint (15i) means that each service worker can only be matched with one elder person or not matched with an elder person.

Model (16) is a bi-objective 0–1 integer programming model that can be solved using a linear weighting method [64]. Model (16) is transformed into the following single objective optimization model:

$$\max Z = \omega_1 \sum_{i=1}^{m} \sum_{j=1}^{n} pr_i p_{ij} x_{ij} + \omega_2 \sum_{i=1}^{m} \sum_{j=1}^{n} q_{ij} x_{ij} \tag{16A}$$

$$s.t. \ (sex_y^i - sex_y^j)x_{ij} \geq 0, y = 1, 2 \tag{16B}$$

$$(l_s^j - l_s^i)x_{ij} \geq 0, s = 1, 2, \ldots, 7 \tag{16C}$$

$$(o_t^j - o_t^i)x_{ij} \geq 0, t = 1, 2, \ldots, 6 \tag{16D}$$

$$(h_r^j - h_r^i)x_{ij} \geq 0, r = 1, 2 \tag{16E}$$

$$(k_v^j - k_v^i)x_{ij} \geq 0, v = 1, 2, \ldots, 6 \tag{16F}$$

$$\sum_{j=1}^{n} x_{ij} \leq 1 \tag{16G}$$

$$\sum_{i=1}^{m} x_{ij} \leq 1 \tag{16H}$$

$$i = 1, 2, \ldots, m; j = 1, 2, \ldots, n \tag{16I}$$

The weights $\omega_1$ and $\omega_2$ reflect the importance of both subjects in the actual matching decision. If $\omega_1 > \omega_2$, it means that the satisfaction degree of the subject on the elderly side should be emphasized in the matching decision; if $\omega_1 < \omega_2$, it means that the satisfaction degree of the subject on the service personnel side should be emphasized in the matching decision; and if $\omega_1 = \omega_2 = 0.5$, it means that the fairness of both subjects should be considered in the matching decision.

The model (15) can be solved using optimization software such as LINGO 18.0, Cplex, Matlab, etc. after being transformed into a single-objective optimization model (16) by the linear weighting method. It should be noted that, following the computation of this model, if an unmatched successful elder remains, that elder is questioned about whether they would like to lessen the requirement of the service personnel's non-elasticity index, or Qualification ($U_7$) indicator requirement. The original optimization model is solved once more if the elder consents to reduce the requirement; if the elder declines, the matching process terminates and some elders are eventually matched.

## 5. Calculations

Assuming that a smart elderly service platform has 10 elderly people to be matched $\{A_1, A_2, A_3, A_4, A_5, A_6, A_7, A_8, A_9, A_{10}\}$, and 15 available elderly service providers $\{B_1, B_2, B_3, B_4, B_5, B_6, B_7, B_8, B_9, B_{10}, B_{11}, B_{12}, B_{13}, B_{14}, B_{15}\}$, the platform starts to match the two parties after collecting their personal information and demand information. In the following, we match these 10 elders with

**Table 1. Expected demand values of elder people for service personnel.**

| Elder $A_i$ | Age $c_1^i$ | Wages (Yuan) $c_2^i$ | Work experience (years) $c_3^i$ | Educational attainmen $c_4^i$ | The sexes $c_5^i$ | Services $c_6^i$ | Qualification $c_7^i$ |
|---|---|---|---|---|---|---|---|
| $A_1$ | [45,50] | 3000 | 8 | 4 | [0 1] | [1 0 1 0 1 1 1] | [1 1 1 1 1 0] |
| $A_2$ | [42,45] | 3500 | 5 | 5 | [1 0] | [1 0 1 0 1 1 1] | [1 1 2 2 1 0] |
| $A_3$ | [50,54] | 2800 | 6 | 5 | [1 1] | [0 0 1 0 0 1 1] | [1 1 1 1 0 1] |
| $A_4$ | [45,50] | 3200 | 5 | 3 | [0 1] | [1 0 1 0 0 1 1] | [1 1 1 2 1 0] |
| $A_5$ | [45,50] | 3500 | 8 | 3 | [0 1] | [1 0 1 0 1 1 1] | [1 1 1 2 1 1] |
| $A_6$ | [40,45] | 3200 | 5 | 3 | [0 1] | [1 1 1 1 1 1 1] | [1 1 1 2 2 1] |
| $A_7$ | [50,55] | 2800 | 8 | 4 | [1 1] | [0 0 1 0 1 1 1] | [1 1 1 1 0 0] |
| $A_8$ | [35,40] | 4000 | 3 | 5 | [1 0] | [1 1 1 1 1 1 1] | [2 2 1 3 2 1] |
| $A_9$ | [50,55] | 3200 | 8 | 3 | [0 1] | [1 0 1 0 1 1 1] | [1 1 1 1 1 1] |
| $A_{10}$ | [38,43] | 3800 | 5 | 5 | [1 1] | [1 1 1 0 1 1 1] | [2 1 1 2 2 1] |

the 15 elderly care service personnel to demonstrate the possible application of the above-proposed method. We also provide a brief overview of the computational procedure and outcomes of using the above-described method to identify the aging-in-place program.

The indicators that elders consider when choosing a service personnel are $U = \{U_1, U_2, U_3, U_4, U_5, U_6, U_7\}$, and these indicators represent, in order, age, salary, work experience, education, gender, job content, and qualifications. The ten elders $\{A_1, A_2, A_3, A_4, A_5, A_6, A_7, A_8, A_9, A_{10}\}$ who will be paired Table 1 displays the service providers' anticipated demands, and Table 2 displays the service providers' actual state in response to those demands.

The indicators considered by the service personnels in selecting elders are $E = \{E_1, E_2, E_3, E_4\}$, and these indicators represent, in order, salary, benefits, form of work, and elders' situation. The expected needs of the 15 available senior service providers $\{B_1, B_2, B_3, B_4, B_5, B_6, B_7, B_8, B_9, B_{10}, B_{11}, B_{12}, B_{13}, B_{14}, B_{15}\}$ are shown in Table 3, and the actual values of the seniors in response to the expected indicators are shown in Table 4.

**Table 2. Actual values of service personnel in response to expectations.**

| Service provider $B_j$ | Age $d_1^j$ | Wages (Yuan) $d_2^j$ | Work experience (years) $d_3^j$ | Educational attainmen $d_4^j$ | The sexes $d_5^j$ | Services $d_6^j$ | Qualification $d_7^j$ |
|---|---|---|---|---|---|---|---|
| $B_1$ | 42 | 4000 | 5 | 5 | [0 1] | [1 1 1 1 1 1 1] | [2 1 1 1 1 0] |
| $B_2$ | 46 | 3500 | 8 | 5 | [0 1] | [1 1 1 0 1 0 1] | [1 1 1 1 1 1] |
| $B_3$ | 45 | 3000 | 6 | 5 | [0 1] | [1 1 1 1 1 1 1] | [2 2 1 3 1 0] |
| $B_4$ | 48 | 3000 | 8 | 4 | [0 1] | [1 1 1 1 1 1 1] | [1 1 1 1 2 0] |
| $B_5$ | 43 | 3500 | 3 | 4 | [0 1] | [1 1 1 1 1 1 1] | [2 1 1 2 1 1] |
| $B_6$ | 42 | 3800 | 3 | 6 | [0 1] | [1 1 1 1 1 1 1] | [2 1 1 3 2 1] |
| $B_7$ | 50 | 3000 | 10 | 3 | [1 0] | [1 1 1 0 1 1 1] | [2 1 2 2 2 1] |
| $B_8$ | 45 | 4200 | 8 | 5 | [1 0] | [1 1 1 1 1 1 1] | [2 1 1 2 1 0] |
| $B_9$ | 55 | 3000 | 12 | 3 | [0 1] | [1 0 1 0 1 0 1] | [1 1 1 1 1 0] |
| $B_{10}$ | 42 | 3800 | 5 | 5 | [0 1] | [1 1 1 1 1 1 1] | [2 1 1 2 2 1] |
| $B_{11}$ | 40 | 4000 | 3 | 5 | [0 1] | [1 1 1 1 1 1 1] | [2 2 1 3 1 0] |
| $B_{12}$ | 51 | 3000 | 6 | 3 | [1 0] | [1 1 1 1 1 1 1] | [2 1 1 1 0 0] |
| $B_{13}$ | 49 | 3500 | 9 | 4 | [0 1] | [1 1 1 1 1 1 1] | [1 1 1 1 1 0] |
| $B_{14}$ | 46 | 3800 | 7 | 4 | [1 0] | [1 1 1 1 1 1 1] | [1 1 1 2 1 1] |
| $B_{15}$ | 52 | 3200 | 10 | 3 | [1 0] | [1 1 1 1 1 1 1] | [2 1 1 1 1 0] |

**Table 3. Expected demand values of service personnel for elders.**

| Service personnel $B_j$ | Wages (Yuan) $f_1^j$ | Welfare treatment $f_2^j$ | Forms of work $f_3^j$ | Situation of the elderly $f_4^j$ |
|---|---|---|---|---|
| $B_1$ | 4000 | $\{1,1,0,1,1,0\}$ | $[1 \quad 1]$ | $[1 \quad 1 \quad 1 \quad 1 \quad 1 \quad 1]$ |
| $B_2$ | 3500 | $\{1,1,1,1,1,0\}$ | $[1 \quad 1]$ | $[1 \quad 1 \quad 1 \quad 1 \quad 0 \quad 1]$ |
| $B_3$ | 3000 | $\{0,1,1,1,1,1\}$ | $[0 \quad 1]$ | $[1 \quad 1 \quad 1 \quad 1 \quad 0 \quad 0]$ |
| $B_4$ | 3000 | $\{0,1,1,1,1,1\}$ | $[1 \quad 1]$ | $[1 \quad 1 \quad 1 \quad 1 \quad 0 \quad 1]$ |
| $B_5$ | 3500 | $\{1,1,1,1,1,1\}$ | $[0 \quad 1]$ | $[1 \quad 1 \quad 1 \quad 1 \quad 1 \quad 1]$ |
| $B_6$ | 3800 | $\{1,1,1,1,1,1\}$ | $[1 \quad 1]$ | $[1 \quad 1 \quad 1 \quad 1 \quad 1 \quad 1]$ |
| $B_7$ | 3000 | $\{1,1,0,1,1,0\}$ | $[0 \quad 1]$ | $[1 \quad 1 \quad 1 \quad 1 \quad 0 \quad 0]$ |
| $B_8$ | 4200 | $\{0,1,1,1,0,1\}$ | $[1 \quad 1]$ | $[1 \quad 1 \quad 1 \quad 1 \quad 1 \quad 1]$ |
| $B_9$ | 3000 | $\{1,1,1,1,1,0\}$ | $[0 \quad 1]$ | $[1 \quad 1 \quad 1 \quad 1 \quad 0 \quad 1]$ |
| $B_{10}$ | 3800 | $\{1,1,0,1,1,1\}$ | $[1 \quad 0]$ | $[1 \quad 1 \quad 1 \quad 1 \quad 1 \quad 1]$ |
| $B_{11}$ | 4000 | $\{1,1,1,1,1,1\}$ | $[1 \quad 1]$ | $[1 \quad 1 \quad 1 \quad 1 \quad 1 \quad 1]$ |
| $B_{12}$ | 3000 | $\{1,1,0,1,1,0\}$ | $[1 \quad 1]$ | $[1 \quad 1 \quad 1 \quad 1 \quad 0 \quad 0]$ |
| $B_{13}$ | 3500 | $\{1,1,1,1,1,1\}$ | $[1 \quad 1]$ | $[1 \quad 1 \quad 1 \quad 1 \quad 1 \quad 1]$ |
| $B_{14}$ | 3800 | $\{1,1,0,1,1,1\}$ | $[1 \quad 1]$ | $[1 \quad 1 \quad 1 \quad 1 \quad 1 \quad 1]$ |
| $B_{15}$ | 3200 | $\{1,1,1,1,1,1\}$ | $[1 \quad 1]$ | $[1 \quad 1 \quad 1 \quad 1 \quad 1 \quad 1]$ |

The satisfaction of the elderly with the elasticity indicators of the service personnel is calculated separately by the above formula for expectation satisfaction, i.e., the satisfaction of the age indicator is shown in Table 5, where $c_{1max} = 55$, $c_{1min} = 35$; the highest wage value that the elderly can give in the wage indicator is $c_{2max}^1 = 4000$, $c_{2max}^2 = 4800$, $c_{2max}^3 = 4000$, $c_{2max}^4 = 4000$, $c_{2max}^5 = 4500$, $c_{2max}^6 = 4200$, $c_{2max}^7 = 3500$, $c_{2max}^8 = 5000$, $c_{2max}^9 = 4200$, $c_{2max}^{10} = 5000$. The satisfaction of the wage indicator is shown in Table 6. The satisfaction level of work experience indicator is shown in Table 7. The satisfaction level of the education level indicator is shown in Table 8. The degree of importance attached by the elderly to each indicator of service personnel is shown in Table 9.

According to Tables 5–9 and Eq (5), the total satisfaction of the elderly with the service personnel was calculated as follows Table 10 Shown.

The minimum wage values accepted by service members are: $f_{1min}^1 = 3000$, $f_{1min}^2 = 2500$, $f_{1min}^3 = 2200$, $f_{1min}^4 = 2500$, $f_{1min}^5 = 2500$, $f_{1min}^6 = 3000$, $f_{1min}^7 = 2500$, $f_{1min}^8 = 3500$, $f_{1min}^9 = 2000$, $f_{1min}^{10} = 2800$, $f_{1min}^{11} = 3000$, $f_{1min}^{12} = 2000$, $f_{1min}^{13} = 2800$, $f_{1min}^{14} = 2800$, $f_{1min}^{15} = 2500$. Table 11 displays the service personnel' satisfaction with wage indicators for the elderly, while Table 12

**Table 4. Actual values of elder's demands in response to expectations.**

| Elder $A_i$ | Wages (Yuan) $g_1^i$ | Welfare treatment $g_2^i$ | Forms of work $g_3^i$ | Situation of the elderly $g_4^i$ |
|---|---|---|---|---|
| $A_1$ | 3000 | $\{0,1,0,1,1,1\}$ | $[0 \quad 1]$ | $[0 \quad 0 \quad 0 \quad 1 \quad 0 \quad 0]$ |
| $A_2$ | 3500 | $\{1,1,0,1,1,1\}$ | $[1 \quad 0]$ | $[0 \quad 0 \quad 0 \quad 1 \quad 0 \quad 0]$ |
| $A_3$ | 2800 | $\{0,1,1,1,1,1\}$ | $[0 \quad 1]$ | $[0 \quad 0 \quad 1 \quad 0 \quad 0 \quad 0]$ |
| $A_4$ | 3200 | $\{1,1,1,1,1,1\}$ | $[0 \quad 1]$ | $[0 \quad 0 \quad 0 \quad 1 \quad 0 \quad 0]$ |
| $A_5$ | 3500 | $\{1,1,1,1,1,1\}$ | $[0 \quad 1]$ | $[0 \quad 0 \quad 0 \quad 1 \quad 0 \quad 0]$ |
| $A_6$ | 3200 | $\{0,1,1,1,1,1\}$ | $[0 \quad 1]$ | $[0 \quad 0 \quad 0 \quad 0 \quad 0 \quad 1]$ |
| $A_7$ | 2800 | $\{1,1,1,1,1,1\}$ | $[0 \quad 1]$ | $[1 \quad 0 \quad 0 \quad 0 \quad 0 \quad 0]$ |
| $A_8$ | 4000 | $\{1,1,1,0,1,1\}$ | $[0 \quad 1]$ | $[0 \quad 0 \quad 0 \quad 0 \quad 0 \quad 1]$ |
| $A_9$ | 3200 | $\{1,1,0,1,0,1\}$ | $[1 \quad 0]$ | $[0 \quad 1 \quad 0 \quad 0 \quad 0 \quad 0]$ |
| $A_{10}$ | 3800 | $\{1,1,0,0,1,1\}$ | $[0 \quad 1]$ | $[0 \quad 0 \quad 0 \quad 0 \quad 1 \quad 0]$ |

**Table 5. Elder people's satisfaction with the age of service personnel.**

| $A_i$ | $p_{i1}^1$ | $p_{i2}^1$ | $p_{i3}^1$ | $p_{i4}^1$ | $p_{i5}^1$ | $p_{i6}^1$ | $p_{i7}^1$ | $p_{i8}^1$ | $p_{i9}^1$ | $p_{i10}^1$ | $p_{i11}^1$ | $p_{i12}^1$ | $p_{i13}^1$ | $p_{i14}^1$ | $p_{i15}^1$ |
|---|---|---|---|---|---|---|---|---|---|---|---|---|---|---|---|
| $A_1$ | 0.70 | 1.00 | 1.00 | 1.00 | 0.80 | 0.70 | 1.00 | 1.00 | 0.00 | 0.70 | 0.50 | 0.80 | 1.00 | 1.00 | 0.60 |
| $A_2$ | 1.00 | 0.90 | 1.00 | 0.70 | 1.00 | 1.00 | 0.50 | 1.00 | 0.00 | 1.00 | 0.71 | 0.40 | 0.60 | 0.90 | 0.30 |
| $A_3$ | 0.47 | 0.73 | 0.67 | 0.87 | 0.53 | 0.47 | 1.00 | 0.67 | 0.00 | 0.47 | 0.33 | 1.00 | 0.93 | 0.73 | 1.00 |
| $A_4$ | 0.70 | 1.00 | 1.00 | 1.00 | 0.80 | 0.70 | 1.00 | 1.00 | 0.00 | 0.70 | 0.50 | 0.80 | 1.00 | 1.00 | 0.60 |
| $A_5$ | 0.70 | 1.00 | 1.00 | 1.00 | 0.80 | 0.70 | 1.00 | 1.00 | 0.00 | 0.70 | 0.50 | 0.80 | 1.00 | 1.00 | 0.60 |
| $A_6$ | 1.00 | 0.90 | 1.00 | 0.70 | 1.00 | 1.00 | 0.50 | 1.00 | 0.00 | 1.00 | 1.00 | 0.40 | 0.60 | 0.90 | 0.30 |
| $A_7$ | 0.47 | 0.73 | 0.67 | 0.87 | 0.53 | 0.47 | 1.00 | 0.67 | 1.00 | 0.47 | 0.33 | 1.00 | 0.93 | 0.73 | 1.00 |
| $A_8$ | 0.87 | 0.60 | 0.67 | 0.47 | 0.80 | 0.87 | 0.33 | 0.67 | 0.00 | 0.87 | 1.00 | 0.27 | 0.40 | 0.60 | 0.20 |
| $A_9$ | 0.47 | 0.73 | 0.67 | 0.87 | 0.53 | 0.47 | 1.00 | 0.67 | 1.00 | 0.47 | 0.33 | 1.00 | 0.93 | 0.73 | 1.00 |
| $A_{10}$ | 1.00 | 0.75 | 0.83 | 0.58 | 1.00 | 1.00 | 0.42 | 0.83 | 0.00 | 1.00 | 1.00 | 0.33 | 0.50 | 0.75 | 0.25 |

displays their satisfaction with welfare benefits. Table 13 shows the service workers' relative importance of each elderly indicator.

Based on Tables 11–13 and Eq (8), the total satisfaction of the service personnel with the elderly was calculated as follows Table 14 Shown.

A multi-objective optimization model (9) can be built based on how well the elderly and service staff are satisfied, and once the optimization software LINGO 18.0 has been used to find a solution, it can be changed to a single-objective model (10). After the calculation of no solution, the constraints (9h) to the constraints $\sum_{j=1}^{n} x_{ij} \leq 1$ after the optimization model for the calculation of the results: $x_{1,4} = 1$, $x_{3,7} = 1$, $x_{4,3} = 1$, $x_{5,5} = 1$, $x_{7,12} = 1$, $x_{9,10} = 1$, $x_{10,6} = 1$, that is, $A_1 \leftrightarrow B_4$, $A_3 \leftrightarrow B_7$, $A_4 \leftrightarrow B_3$, $A_5 \leftrightarrow B_5$, $A_7 \leftrightarrow B_{12}$, $A_9 \leftrightarrow B_{10}$, $A_{10} \leftrightarrow B_6$, the elderly $A_2$, $A_6$, $A_8$ did not match to the service personnel. One of them, the elderly $A_2$, $A_6$, has non-elastic expectation indicator constraints that are too high, meaning there are no service personnel to meet the requirements. The elderly $A_8$, due to service resource conflicts, matches the same service personnel with non-elastic indicator requirements. After questioning the elderly $A_2$, $A_6$ and $A_8$, the elderly decided not to lower the requirement of the service personnel qualification index. We must take into consideration the priority of these ten elderly individuals, whose basic index information is as follows, Table 15. According to the Formulas (11)–(14) the calculated priority value: $pr_1 = 0.55$, $pr_2 = 0.37$, $pr_3 = 0.58$, $pr_4 = 0.62$, $pr_5 = 0.25$, $pr_6 = 0.97$, $pr_7 = 0.25$, $pr_8 = 0.67$, $pr_9 = 0.17$, $pr_{10} = 0.58$.

A multi-objective optimization model (15) can be created, transformed into a single-objective model (16), and then solved using the optimization software LINGO 18.0, taking into

**Table 6. Elderly people's satisfaction with service personnel' wages.**

| $A_i$ | $p_{i1}^2$ | $p_{i2}^2$ | $p_{i3}^2$ | $p_{i4}^2$ | $p_{i5}^2$ | $p_{i6}^2$ | $p_{i7}^2$ | $p_{i8}^2$ | $p_{i9}^2$ | $p_{i10}^2$ | $p_{i11}^2$ | $p_{i12}^2$ | $p_{i13}^2$ | $p_{i14}^2$ | $p_{i15}^2$ |
|---|---|---|---|---|---|---|---|---|---|---|---|---|---|---|---|
| $A_1$ | 0.00 | 0.62 | 0.00 | 0.00 | 0.50 | 0.20 | 0.00 | 1.00 | 0.20 | 0.83 | 0.00 | 0.62 | 0.00 | 0.00 | 0.50 |
| $A_2$ | 0.50 | 1.00 | 0.42 | 0.63 | 1.00 | 0.70 | 0.00 | 1.00 | 0.70 | 1.00 | 0.50 | 1.00 | 0.42 | 0.63 | 1.00 |
| $A_3$ | 1.00 | 1.00 | 0.83 | 1.00 | 1.00 | 1.00 | 0.71 | 1.00 | 1.00 | 1.00 | 1.00 | 1.00 | 0.83 | 1.00 | 1.00 |
| $A_4$ | 1.00 | 1.00 | 0.83 | 1.00 | 1.00 | 1.00 | 0.71 | 1.00 | 1.00 | 1.00 | 1.00 | 1.00 | 0.83 | 1.00 | 1.00 |
| $A_5$ | 0.50 | 1.00 | 0.42 | 0.63 | 1.00 | 0.70 | 0.00 | 1.00 | 0.70 | 1.00 | 0.50 | 1.00 | 0.42 | 0.63 | 1.00 |
| $A_6$ | 0.20 | 0.77 | 0.17 | 0.25 | 0.70 | 0.40 | 0.00 | 1.00 | 0.40 | 1.00 | 0.20 | 0.77 | 0.17 | 0.25 | 0.70 |
| $A_7$ | 1.00 | 1.00 | 0.83 | 1.00 | 1.00 | 1.00 | 0.71 | 1.00 | 1.00 | 1.00 | 1.00 | 1.00 | 0.83 | 1.00 | 1.00 |
| $A_8$ | 0.00 | 0.46 | 0.00 | 0.00 | 0.30 | 0.00 | 0.00 | 0.80 | 0.00 | 0.67 | 0.00 | 0.46 | 0.00 | 0.00 | 0.30 |
| $A_9$ | 1.00 | 1.00 | 0.83 | 1.00 | 1.00 | 1.00 | 0.71 | 1.00 | 1.00 | 1.00 | 1.00 | 1.00 | 0.83 | 1.00 | 1.00 |
| $A_{10}$ | 0.20 | 0.77 | 0.17 | 0.25 | 0.70 | 0.40 | 0.00 | 1.00 | 0.40 | 1.00 | 0.20 | 0.77 | 0.17 | 0.25 | 0.70 |

**Table 7. Elder people's satisfaction with the work experience of service personnel.**

| $A_i$ | $p_{i1}^3$ | $p_{i2}^3$ | $p_{i3}^3$ | $p_{i4}^3$ | $p_{i5}^3$ | $p_{i6}^3$ | $p_{i7}^3$ | $p_{i8}^3$ | $p_{i9}^3$ | $p_{i10}^3$ | $p_{i11}^3$ | $p_{i12}^3$ | $p_{i13}^3$ | $p_{i14}^3$ | $p_{i15}^3$ |
|---|---|---|---|---|---|---|---|---|---|---|---|---|---|---|---|
| $A_1$ | 0.63 | 1.00 | 0.75 | 1.00 | 0.38 | 0.38 | 1.00 | 1.00 | 1.00 | 0.63 | 0.38 | 0.75 | 1.00 | 0.88 | 1.00 |
| $A_2$ | 1.00 | 1.00 | 1.00 | 1.00 | 0.60 | 0.60 | 1.00 | 1.00 | 1.00 | 1.00 | 0.60 | 1.00 | 1.00 | 1.00 | 1.00 |
| $A_3$ | 0.83 | 1.00 | 1.00 | 1.00 | 0.50 | 0.50 | 1.00 | 1.00 | 2.00 | 0.83 | 0.50 | 1.00 | 1.00 | 1.00 | 1.00 |
| $A_4$ | 1.00 | 1.00 | 1.00 | 1.00 | 0.60 | 0.60 | 1.00 | 1.00 | 1.00 | 1.00 | 0.60 | 1.00 | 1.00 | 1.00 | 1.00 |
| $A_5$ | 0.63 | 1.00 | 0.75 | 1.00 | 0.38 | 0.38 | 1.00 | 1.00 | 1.00 | 0.63 | 0.38 | 0.75 | 1.00 | 0.88 | 1.00 |
| $A_6$ | 1.00 | 1.00 | 1.00 | 1.00 | 0.60 | 0.60 | 1.00 | 1.00 | 1.00 | 1.00 | 0.60 | 1.00 | 1.00 | 1.00 | 1.00 |
| $A_7$ | 0.63 | 1.00 | 0.75 | 1.00 | 0.38 | 0.38 | 1.00 | 1.00 | 1.00 | 0.63 | 0.38 | 0.75 | 1.00 | 0.88 | 1.00 |
| $A_8$ | 1.00 | 1.00 | 1.00 | 1.00 | 1.00 | 1.00 | 1.00 | 1.00 | 1.00 | 1.00 | 1.00 | 1.00 | 1.00 | 1.00 | 1.00 |
| $A_9$ | 0.63 | 1.00 | 0.75 | 1.00 | 0.38 | 0.38 | 1.00 | 1.00 | 1.00 | 0.63 | 0.38 | 0.75 | 1.00 | 0.88 | 1.00 |
| $A_{10}$ | 1.00 | 1.00 | 1.00 | 1.00 | 0.60 | 0.60 | 1.00 | 1.00 | 1.00 | 1.00 | 0.60 | 1.00 | 1.00 | 1.00 | 1.00 |

account the elderly's priorities and level of satisfaction with service personnel. The matching results are obtained as follows: $x_{14} = 1$, $x_{37} = 1$, $x_{43} = 1$, $x_{55} = 1$, $x_{66} = 1$, $x_{712} = 1$, $x_{910} = 1$, i.e., $A_1 \leftrightarrow B_4$, $A_3 \leftrightarrow B_7$, $A_4 \leftrightarrow B_3$, $A_5 \leftrightarrow B_5$, $A_6 \leftrightarrow B_6$, $A_7 \leftrightarrow B_{12}$, $A_9 \leftrightarrow B_{10}$. The elderly $A_2$, $A_8$, and $A_{10}$ and did not correspond with the appropriate and qualified service personnel. Elders both $A_2$ and $A_8$ have decided not to lower the standards of the service personnel qualification index. Then ask elderly people $A_{10}$ after choosing to reduce the requirements of the service personnel qualification index, the qualification index of elderly people $A_{10}$ indicator matrix $c_7^{10} = \begin{bmatrix} 2 & 1 & 1 & 2 & 2 & 1 \end{bmatrix}$ changed to $c_7'10 = \begin{bmatrix} 1 & 1 & 1 & 2 & 1 & 1 \end{bmatrix}$. Recalculate the model's calculations while accounting for its priority. The model's solution will be obtained by turning it into a single-objective model using the optimization program LINGO 18.0. The matching results obtained as follows: $x_{1,4} = 1$, $x_{3,7} = 1$, $x_{4,3} = 1$, $x_{5,5} = 1$, $x_{6,6} = 1$, $x_{7,12} = 1$, $x_{9,10} = 1$, $x_{10,14} = 1$, i.e., $A_1 \leftrightarrow B_4$, $A_3 \leftrightarrow B_7$, $A_4 \leftrightarrow B_3$, $A_5 \leftrightarrow B_5$, $A_6 \leftrightarrow B_6$, $A_7 \leftrightarrow B_{12}$, $A_9 \leftrightarrow B_{10}$, $A_{10} \leftrightarrow B_{14}$, the elderly $A_2$ and $A_8$ does not match with the service provider who is suitable and meets the requirements. Then, according to the expectation and demand of the elderly, we can find two temporary old-age service personnel to meet the needs of the elderly, so that the elderly with the needs of the elderly can be matched with the appropriate service personnel.

## 6. Discuss

The bilateral matching problem between senior service personnel and older clients cannot be successfully solved by traditional methods since they are less tailored to the unique needs of the elderly population and typically only take into account the preferences or overall level of

**Table 8. Elderly people's satisfaction with the level of education of service personnel.**

| $A_i$ | $p_{i1}^4$ | $p_{i2}^4$ | $p_{i3}^4$ | $p_{i4}^4$ | $p_{i5}^4$ | $p_{i6}^4$ | $p_{i7}^4$ | $p_{i8}^4$ | $p_{i9}^4$ | $p_{i10}^4$ | $p_{i11}^4$ | $p_{i12}^4$ | $p_{i13}^4$ | $p_{i14}^4$ | $p_{i15}^4$ |
|---|---|---|---|---|---|---|---|---|---|---|---|---|---|---|---|
| $A_1$ | 1.00 | 1.00 | 1.00 | 1.00 | 1.00 | 1.00 | 0.75 | 1.00 | 0.75 | 1.00 | 1.00 | 0.75 | 1.00 | 1.00 | 0.75 |
| $A_2$ | 1.00 | 1.00 | 1.00 | 1.00 | 0.60 | 0.60 | 1.00 | 1.00 | 1.00 | 1.00 | 0.60 | 1.00 | 1.00 | 1.00 | 1.00 |
| $A_3$ | 0.83 | 1.00 | 1.00 | 1.00 | 0.50 | 0.50 | 1.00 | 1.00 | 1.00 | 0.83 | 0.50 | 1.00 | 1.00 | 1.00 | 1.00 |
| $A_4$ | 1.00 | 1.00 | 1.00 | 1.00 | 0.60 | 0.60 | 1.00 | 1.00 | 1.00 | 1.00 | 0.60 | 1.00 | 1.00 | 1.00 | 1.00 |
| $A_5$ | 0.63 | 1.00 | 0.75 | 1.00 | 0.38 | 0.38 | 1.00 | 1.00 | 1.00 | 0.63 | 0.38 | 0.75 | 1.00 | 0.88 | 1.00 |
| $A_6$ | 1.00 | 1.00 | 1.00 | 1.00 | 0.60 | 0.60 | 1.00 | 1.00 | 1.00 | 1.00 | 0.60 | 1.00 | 1.00 | 1.00 | 1.00 |
| $A_7$ | 0.63 | 1.00 | 0.75 | 1.00 | 0.38 | 0.38 | 1.00 | 1.00 | 1.00 | 0.63 | 0.38 | 0.75 | 1.00 | 0.88 | 1.00 |
| $A_8$ | 1.00 | 1.00 | 1.00 | 1.00 | 1.00 | 1.00 | 1.00 | 1.00 | 1.00 | 1.00 | 1.00 | 1.00 | 1.00 | 1.00 | 1.00 |
| $A_9$ | 0.63 | 1.00 | 0.75 | 1.00 | 0.38 | 0.38 | 1.00 | 1.00 | 1.00 | 0.63 | 0.38 | 0.75 | 1.00 | 0.88 | 1.00 |
| $A_{10}$ | 1.00 | 1.00 | 1.00 | 1.00 | 0.60 | 0.60 | 1.00 | 1.00 | 1.00 | 1.00 | 0.60 | 1.00 | 1.00 | 1.00 | 1.00 |

**Table 9. Level of importance attached to each indicator by the elderly.**

| $A_i$ | $w_i^1$ | $w_i^2$ | $w_i^3$ | $w_i^4$ |
|---|---|---|---|---|
| $A_1$ | 0.4 | 0.2 | 0.2 | 0.2 |
| $A_2$ | 0.4 | 0.2 | 0.2 | 0.2 |
| $A_3$ | 0.3 | 0.3 | 0.2 | 0.2 |
| $A_4$ | 0.3 | 0.3 | 0.2 | 0.2 |
| $A_5$ | 0.4 | 0.2 | 0.2 | 0.2 |
| $A_6$ | 0.3 | 0.3 | 0.2 | 0.2 |
| $A_7$ | 0.4 | 0.2 | 0.2 | 0.2 |
| $A_8$ | 0.4 | 0.2 | 0.2 | 0.2 |
| $A_9$ | 0.4 | 0.2 | 0.2 | 0.2 |
| $A_{10}$ | 0.3 | 0.3 | 0.2 | 0.2 |

**Table 10. Elder people's overall satisfaction with service personnel.**

| $A_i$ | $p_{i1}$ | $p_{i2}$ | $p_{i3}$ | $p_{i4}$ | $p_{i5}$ | $p_{i6}$ | $p_{i7}$ | $p_{i8}$ | $p_{i9}$ | $p_{i10}$ | $p_{i11}$ | $p_{i12}$ | $p_{i13}$ | $p_{i14}$ | $p_{i15}$ |
|---|---|---|---|---|---|---|---|---|---|---|---|---|---|---|---|
| $A_1$ | 0.61 | 0.90 | 0.95 | 1.00 | 0.70 | 0.60 | 0.95 | 0.80 | 0.55 | 0.65 | 0.48 | 0.82 | 0.90 | 0.82 | 0.75 |
| $A_2$ | 0.92 | 0.96 | 1.00 | 0.88 | 0.84 | 0.79 | 0.80 | 0.89 | 0.60 | 0.95 | 0.65 | 0.76 | 0.84 | 0.91 | 0.72 |
| $A_3$ | 0.47 | 0.74 | 0.85 | 0.91 | 0.48 | 0.39 | 0.95 | 0.60 | 0.65 | 0.52 | 0.30 | 0.95 | 0.80 | 0.67 | 0.90 |
| $A_4$ | 0.61 | 0.89 | 1.00 | 1.00 | 0.67 | 0.53 | 1.00 | 0.70 | 0.70 | 0.69 | 0.39 | 0.94 | 0.89 | 0.78 | 0.88 |
| $A_5$ | 0.63 | 1.00 | 0.90 | 1.00 | 0.67 | 0.57 | 1.00 | 0.86 | 0.60 | 0.67 | 0.45 | 0.82 | 1.00 | 0.89 | 0.84 |
| $A_6$ | 0.76 | 0.88 | 1.00 | 0.91 | 0.75 | 0.66 | 0.85 | 0.70 | 0.70 | 0.82 | 0.60 | 0.82 | 0.79 | 0.79 | 0.79 |
| $A_7$ | 0.44 | 0.69 | 0.71 | 0.89 | 0.36 | 0.34 | 0.94 | 0.67 | 0.94 | 0.44 | 0.28 | 0.84 | 0.77 | 0.64 | 0.89 |
| $A_8$ | 0.95 | 0.84 | 0.87 | 0.79 | 0.92 | 0.95 | 0.73 | 0.83 | 0.60 | 0.95 | 1.00 | 0.71 | 0.76 | 0.84 | 0.68 |
| $A_9$ | 0.48 | 0.83 | 0.77 | 0.95 | 0.50 | 0.42 | 1.00 | 0.67 | 1.00 | 0.52 | 0.32 | 0.90 | 0.91 | 0.72 | 1.00 |
| $A_{10}$ | 0.95 | 0.93 | 0.95 | 0.87 | 0.84 | 0.84 | 0.83 | 0.85 | 0.70 | 1.00 | 0.79 | 0.80 | 0.85 | 0.93 | 0.78 |

**Table 11. Service personnel satisfaction with elderly wages.**

| $B_j$ | $q_{1j}^1$ | $q_{2j}^1$ | $q_{3j}^1$ | $q_{4j}^1$ | $q_{5j}^1$ | $q_{6j}^1$ | $q_{7j}^1$ | $q_{8j}^1$ | $q_{9j}^1$ | $q_{10j}^1$ |
|---|---|---|---|---|---|---|---|---|---|---|
| $B_1$ | 0.00 | 0.50 | 0.00 | 0.20 | 0.50 | 0.20 | 0.00 | 1.00 | 0.20 | 0.80 |
| $B_2$ | 0.50 | 1.00 | 0.30 | 0.70 | 1.00 | 0.70 | 0.30 | 1.00 | 0.70 | 1.00 |
| $B_3$ | 1.00 | 1.00 | 0.75 | 1.00 | 1.00 | 1.00 | 0.75 | 1.00 | 1.00 | 1.00 |
| $B_4$ | 1.00 | 1.00 | 0.60 | 1.00 | 1.00 | 1.00 | 0.60 | 1.00 | 1.00 | 1.00 |
| $B_5$ | 1.00 | 1.00 | 0.60 | 1.00 | 1.00 | 1.00 | 0.60 | 1.00 | 1.00 | 1.00 |
| $B_6$ | 0.50 | 1.00 | 0.30 | 0.70 | 1.00 | 0.70 | 0.30 | 1.00 | 0.70 | 1.00 |
| $B_7$ | 1.00 | 1.00 | 0.60 | 1.00 | 1.00 | 1.00 | 0.60 | 1.00 | 1.00 | 1.00 |
| $B_8$ | 0.00 | 0.00 | 0.00 | 0.00 | 0.00 | 0.00 | 0.00 | 0.71 | 0.00 | 0.43 |
| $B_9$ | 1.00 | 1.00 | 0.80 | 1.00 | 1.00 | 1.00 | 0.80 | 1.00 | 1.00 | 1.00 |
| $B_{10}$ | 0.20 | 0.70 | 0.00 | 0.40 | 0.70 | 0.40 | 0.00 | 1.00 | 0.40 | 1.00 |
| $B_{11}$ | 0.00 | 0.50 | 0.00 | 0.20 | 0.50 | 0.20 | 0.00 | 1.00 | 0.20 | 0.80 |
| $B_{12}$ | 1.00 | 1.00 | 0.80 | 1.00 | 1.00 | 1.00 | 0.80 | 1.00 | 1.00 | 1.00 |
| $B_{13}$ | 0.29 | 1.00 | 0.00 | 0.57 | 1.00 | 0.57 | 0.00 | 1.00 | 0.57 | 1.00 |
| $B_{14}$ | 0.20 | 0.70 | 0.00 | 0.40 | 0.70 | 0.40 | 0.00 | 1.00 | 0.40 | 1.00 |
| $B_{15}$ | 0.71 | 1.00 | 0.43 | 1.00 | 1.00 | 1.00 | 0.43 | 1.00 | 1.00 | 1.00 |

**Table 12. Satisfaction of service personnel with welfare benefits for the elderly.**

| $B_j$ | $q_{1j}^2$ | $q_{2j}^2$ | $q_{3j}^2$ | $q_{4j}^2$ | $q_{5j}^2$ | $q_{6j}^2$ | $q_{7j}^2$ | $q_{8j}^2$ | $q_{9j}^2$ | $q_{10j}^2$ |
|---|---|---|---|---|---|---|---|---|---|---|
| $B_1$ | 0.75 | 1.00 | 0.75 | 1.00 | 1.00 | 0.75 | 1.00 | 0.75 | 0.75 | 0.75 |
| $B_2$ | 0.60 | 0.80 | 0.80 | 1.00 | 1.00 | 0.80 | 1.00 | 0.80 | 0.60 | 0.60 |
| $B_3$ | 0.80 | 0.80 | 1.00 | 1.00 | 1.00 | 1.00 | 1.00 | 0.80 | 0.60 | 0.60 |
| $B_4$ | 0.80 | 0.80 | 1.00 | 1.00 | 1.00 | 1.00 | 1.00 | 0.80 | 0.60 | 0.60 |
| $B_5$ | 0.67 | 0.83 | 0.83 | 1.00 | 1.00 | 0.83 | 1.00 | 0.83 | 0.67 | 0.67 |
| $B_6$ | 0.67 | 0.83 | 0.83 | 1.00 | 1.00 | 0.83 | 1.00 | 0.83 | 0.67 | 0.67 |
| $B_7$ | 0.75 | 1.00 | 0.75 | 1.00 | 1.00 | 0.75 | 1.00 | 0.75 | 0.75 | 0.75 |
| $B_8$ | 0.75 | 0.75 | 1.00 | 1.00 | 1.00 | 1.00 | 1.00 | 0.75 | 0.75 | 0.50 |
| $B_9$ | 0.60 | 0.80 | 0.80 | 1.00 | 1.00 | 0.80 | 1.00 | 0.80 | 0.60 | 0.60 |
| $B_{10}$ | 0.80 | 1.00 | 0.80 | 0.80 | 1.00 | 1.00 | 1.00 | 0.80 | 0.80 | 0.80 |
| $B_{11}$ | 0.67 | 0.83 | 0.83 | 1.00 | 1.00 | 0.83 | 1.00 | 0.83 | 0.67 | 0.67 |
| $B_{12}$ | 0.75 | 1.00 | 0.75 | 1.00 | 1.00 | 0.75 | 1.00 | 0.75 | 0.75 | 0.75 |
| $B_{13}$ | 0.67 | 0.83 | 0.83 | 1.00 | 1.00 | 0.83 | 1.00 | 0.83 | 0.67 | 0.67 |
| $B_{14}$ | 0.80 | 1.00 | 0.80 | 0.80 | 1.00 | 1.00 | 1.00 | 0.80 | 0.80 | 0.80 |
| $B_{15}$ | 0.67 | 0.83 | 0.83 | 1.00 | 1.00 | 0.83 | 1.00 | 0.83 | 0.67 | 0.67 |

satisfaction of both parties [65, 66]. In this paper, we primarily address the method of matching elderly service personnel, which takes into account various service expectations and the priorities of the elderly in various service resource scenarios. The traditional matching problem takes into account the total satisfaction objective function of both parties when evaluating the satisfaction of a single service expectation or the overall preference of both parties. In contrast, this paper takes into account a variety of service expectations and priorities for the elderly in light of their unique service needs and pays particular attention to human nature, which simultaneously ensures the quality of the service provided by service personnel and prevents the wastage of resources while also taking into account the urgency of the elderly's demand for the service. In addition to guaranteeing the efficiency of service staff and preventing resource waste, it also takes into account the elderly population's pressing need for services. As a result, we contrast these goals with those of the classic bilateral matching problem: the first considers

**Table 13. Level of importance attached to each indicator by service personnel.**

| $B_j$ | $w_j^1$ | $w_j^2$ |
|---|---|---|
| $B_1$ | 0.6 | 0.4 |
| $B_2$ | 0.6 | 0.4 |
| $B_3$ | 0.6 | 0.4 |
| $B_4$ | 0.6 | 0.4 |
| $B_5$ | 0.7 | 0.3 |
| $B_6$ | 0.7 | 0.3 |
| $B_7$ | 0.7 | 0.3 |
| $B_8$ | 0.7 | 0.3 |
| $B_9$ | 0.7 | 0.3 |
| $B_{10}$ | 0.7 | 0.3 |
| $B_{11}$ | 0.7 | 0.3 |
| $B_{12}$ | 0.6 | 0.4 |
| $B_{13}$ | 0.6 | 0.4 |
| $B_{14}$ | 0.6 | 0.4 |
| $B_{15}$ | 0.6 | 0.4 |

**Table 14. Overall satisfaction of service personnel with elder persons.**

| $B_j$ | $q_{1j}$ | $q_{2j}$ | $q_{3j}$ | $q_{4j}$ | $q_{5j}$ | $q_{6j}$ | $q_{7j}$ | $q_{8j}$ | $q_{9j}$ | $q_{10j}$ |
|---|---|---|---|---|---|---|---|---|---|---|
| $B_1$ | 0.30 | 0.70 | 0.30 | 0.52 | 0.65 | 0.37 | 0.40 | 0.90 | 0.42 | 0.78 |
| $B_2$ | 0.54 | 0.92 | 0.50 | 0.82 | 1.00 | 0.73 | 0.58 | 0.92 | 0.66 | 0.84 |
| $B_3$ | 0.92 | 0.92 | 0.85 | 1.00 | 1.00 | 1.00 | 0.85 | 0.92 | 0.84 | 0.84 |
| $B_4$ | 0.92 | 0.92 | 0.76 | 1.00 | 1.00 | 1.00 | 0.76 | 0.92 | 0.84 | 0.84 |
| $B_5$ | 0.87 | 0.93 | 0.69 | 1.00 | 1.00 | 0.95 | 0.76 | 0.93 | 0.87 | 0.87 |
| $B_6$ | 0.57 | 0.93 | 0.51 | 0.82 | 1.00 | 0.74 | 0.58 | 0.93 | 0.69 | 0.87 |
| $B_7$ | 0.90 | 1.00 | 0.66 | 1.00 | 1.00 | 0.93 | 0.76 | 0.90 | 0.90 | 0.90 |
| $B_8$ | 0.30 | 0.30 | 0.40 | 0.40 | 0.30 | 0.30 | 0.40 | 0.73 | 0.30 | 0.46 |
| $B_9$ | 0.84 | 0.92 | 0.80 | 1.00 | 1.00 | 0.94 | 0.88 | 0.92 | 0.84 | 0.84 |
| $B_{10}$ | 0.44 | 0.82 | 0.32 | 0.56 | 0.79 | 0.58 | 0.40 | 0.92 | 0.56 | 0.92 |
| $B_{11}$ | 0.27 | 0.63 | 0.33 | 0.52 | 0.65 | 0.39 | 0.40 | 0.93 | 0.39 | 0.75 |
| $B_{12}$ | 0.90 | 1.00 | 0.78 | 1.00 | 1.00 | 0.93 | 0.88 | 0.90 | 0.90 | 0.90 |
| $B_{13}$ | 0.44 | 0.93 | 0.33 | 0.74 | 1.00 | 0.65 | 0.40 | 0.93 | 0.61 | 0.87 |
| $B_{14}$ | 0.44 | 0.82 | 0.32 | 0.56 | 0.79 | 0.58 | 0.40 | 1.00 | 0.56 | 0.92 |
| $B_{15}$ | 0.70 | 0.93 | 0.59 | 1.00 | 1.00 | 0.95 | 0.66 | 0.93 | 0.87 | 0.87 |

only the elasticity index, while the second considers the maximum amount of satisfaction overall.

Here we transform the inelastic indicators of gender $U_5$, job content $U_6$, qualifications $U_7$, form of work $E_3$, and elderly status $E_4$ into elastic indicators as indicators of satisfaction for the calculation of traditional bilateral matching.

Let $c_5^i = \begin{bmatrix} sex_1^i & sex_2^i \end{bmatrix}$ denote the expected value matrix of the $i$th elder person's choice of service personnel for the indicator $U_5$, and d denote the actual value matrix of the $j$th service personnel for the indicator $U_5$, $sex_y^i, sex_y^j \in \{0, 1\}$, $y = 1,2$. Let $p_{ij}^5$ denote the satisfaction of the $i$th elder person with the gender indicator of the $i$th service personnel, and the formula of $p_{ij}^5$ is shown in Eq (17), and the calculation results are shown in Table 16.

$$p_{ij}^5 = \begin{cases} 1, c_5^i \leq d_5^j \\ 0, c_5^i > d_5^j \end{cases} \tag{17}$$

Let $c_6^i = \begin{bmatrix} l_1^i & l_2^i & l_3^i & l_4^i & l_5^i & l_6^i & l_7^i \end{bmatrix}$ denote the matrix of values of the service personnel's work content indicators $U_6$ for the $i$th elder person, and $d_6^j = \begin{bmatrix} l_1^j & l_2^j & l_3^j & l_4^j & l_5^j & l_6^j & l_7^j \end{bmatrix}$

**Table 15. Information on basic indicators for elders.**

| Elder $A_i$ | Age | Situation of the elderly | Residence |
|---|---|---|---|
| $A_1$ | 68 | $\begin{bmatrix} 0 & 0 & 0 & 1 & 0 & 0 \end{bmatrix}$ | Living alone and not in the same city as their children |
| $A_2$ | 67 | $\begin{bmatrix} 0 & 0 & 0 & 1 & 0 & 0 \end{bmatrix}$ | Living with spouse and not living in the same city as children |
| $A_3$ | 75 | $\begin{bmatrix} 0 & 0 & 1 & 0 & 0 & 0 \end{bmatrix}$ | Living alone and in the same city as children |
| $A_4$ | 72 | $\begin{bmatrix} 0 & 0 & 0 & 1 & 0 & 0 \end{bmatrix}$ | Living alone and not in the same city as their children |
| $A_5$ | 65 | $\begin{bmatrix} 0 & 0 & 0 & 1 & 0 & 0 \end{bmatrix}$ | Living with spouse and in the same city as children |
| $A_6$ | 83 | $\begin{bmatrix} 0 & 0 & 0 & 0 & 0 & 1 \end{bmatrix}$ | Living alone and not in the same city as their children |
| $A_7$ | 65 | $\begin{bmatrix} 1 & 0 & 0 & 0 & 0 & 0 \end{bmatrix}$ | Living alone and in the same city as children |
| $A_8$ | 85 | $\begin{bmatrix} 0 & 0 & 0 & 0 & 0 & 1 \end{bmatrix}$ | Living with children |
| $A_9$ | 70 | $\begin{bmatrix} 0 & 1 & 0 & 0 & 0 & 0 \end{bmatrix}$ | Living with spouse and in the same city as children |
| $A_{10}$ | 70 | $\begin{bmatrix} 0 & 0 & 0 & 0 & 1 & 0 \end{bmatrix}$ | Living with spouse and not living in the same city as children |

**Table 16. Elderly people's satisfaction with the gender of service personnel.**

| $A_i$ | $p_{i1}^5$ | $p_{i2}^5$ | $p_{i3}^5$ | $p_{i4}^5$ | $p_{i5}^5$ | $p_{i6}^5$ | $p_{i7}^5$ | $p_{i8}^5$ | $p_{i9}^5$ | $p_{i10}^5$ | $p_{i11}^5$ | $p_{i12}^5$ | $p_{i13}^5$ | $p_{i14}^5$ | $p_{i15}^5$ |
|---|---|---|---|---|---|---|---|---|---|---|---|---|---|---|---|
| $A_1$ | 1.00 | 1.00 | 1.00 | 1.00 | 1.00 | 1.00 | 0.00 | 0.00 | 1.00 | 1.00 | 1.00 | 0.00 | 1.00 | 0.00 | 0.00 |
| $A_2$ | 0.00 | 0.00 | 0.00 | 0.00 | 0.00 | 0.00 | 1.00 | 1.00 | 0.00 | 0.00 | 0.00 | 1.00 | 0.00 | 1.00 | 1.00 |
| $A_3$ | 1.00 | 1.00 | 1.00 | 1.00 | 1.00 | 1.00 | 1.00 | 1.00 | 1.00 | 1.00 | 1.00 | 1.00 | 1.00 | 1.00 | 1.00 |
| $A_4$ | 1.00 | 1.00 | 1.00 | 1.00 | 1.00 | 1.00 | 0.00 | 0.00 | 1.00 | 1.00 | 1.00 | 0.00 | 1.00 | 0.00 | 0.00 |
| $A_5$ | 1.00 | 1.00 | 1.00 | 1.00 | 1.00 | 1.00 | 0.00 | 0.00 | 1.00 | 1.00 | 1.00 | 0.00 | 1.00 | 0.00 | 0.00 |
| $A_6$ | 1.00 | 1.00 | 1.00 | 1.00 | 1.00 | 1.00 | 0.00 | 0.00 | 1.00 | 1.00 | 1.00 | 0.00 | 1.00 | 0.00 | 0.00 |
| $A_7$ | 1.00 | 1.00 | 1.00 | 1.00 | 1.00 | 1.00 | 1.00 | 1.00 | 1.00 | 1.00 | 1.00 | 1.00 | 1.00 | 1.00 | 1.00 |
| $A_8$ | 0.00 | 0.00 | 0.00 | 0.00 | 0.00 | 0.00 | 1.00 | 1.00 | 0.00 | 0.00 | 0.00 | 1.00 | 0.00 | 1.00 | 1.00 |
| $A_9$ | 1.00 | 1.00 | 1.00 | 1.00 | 1.00 | 1.00 | 0.00 | 0.00 | 1.00 | 1.00 | 1.00 | 0.00 | 1.00 | 0.00 | 0.00 |
| $A_{10}$ | 1.00 | 1.00 | 1.00 | 1.00 | 1.00 | 1.00 | 1.00 | 1.00 | 1.00 | 1.00 | 1.00 | 1.00 | 1.00 | 1.00 | 1.00 |

denote the matrix of values of the service personnel's actual work content indicators for the $j$th service personnel, and $l_s^i, l_s^j \in \{0,1\}$, $s = 1,2,\ldots,7$. Denote $p_{ij}^6$ denotes the satisfaction of the $i$th elder person with the $j$th service personnel's work content indicator $U_6$. The formula for $p_{ij}^6$ is shown in Eq (18), and the results of the calculation are shown in Table 17.

$$p_{ij}^6 = \begin{cases} 1, c_6^i \leq d_6^j \\ 0, c_6^i > d_6^j \end{cases} \tag{18}$$

Let $c_7^i = \begin{bmatrix} o_1^i & o_2^i & o_3^i & o_4^i & o_5^i & o_6^i \end{bmatrix}$ denote the value of the $i$th elder person's indicator of the service personnel's qualification matrix, and $d_7^j = \begin{bmatrix} o_1^j & o_2^j & o_3^j & o_4^j & o_5^j & o_6^j \end{bmatrix}$ denote the value of the $j$th service personnel's indicator of the actual qualification matrix. Let $p_{ij}^7$ denote the satisfaction of the $i$th elder person with the $j$th service personnel's qualification index $U_7$. The formula for $p_{ij}^7$ is shown in Eq (19), and the results of the calculation are shown in Table 18.

$$p_{ij}^7 = \begin{cases} 1, c_7^i \leq d_7^j \\ 0, c_7^i > d_7^j \end{cases} \tag{19}$$

**Table 17. Elderly people's satisfaction with the content of service staff's work.**

| $A_i$ | $p_{i1}^6$ | $p_{i2}^6$ | $p_{i3}^6$ | $p_{i4}^6$ | $p_{i5}^6$ | $p_{i6}^6$ | $p_{i7}^6$ | $p_{i8}^6$ | $p_{i9}^6$ | $p_{i10}^6$ | $p_{i11}^6$ | $p_{i12}^6$ | $p_{i13}^6$ | $p_{i14}^6$ | $p_{i15}^6$ |
|---|---|---|---|---|---|---|---|---|---|---|---|---|---|---|---|
| $A_1$ | 1.00 | 0.00 | 1.00 | 1.00 | 1.00 | 1.00 | 1.00 | 1.00 | 0.00 | 1.00 | 1.00 | 1.00 | 1.00 | 1.00 | 1.00 |
| $A_2$ | 1.00 | 0.00 | 1.00 | 1.00 | 1.00 | 0.00 | 1.00 | 1.00 | 0.00 | 1.00 | 1.00 | 1.00 | 1.00 | 1.00 | 1.00 |
| $A_3$ | 1.00 | 0.00 | 1.00 | 1.00 | 1.00 | 1.00 | 1.00 | 1.00 | 0.00 | 1.00 | 1.00 | 1.00 | 1.00 | 1.00 | 1.00 |
| $A_4$ | 1.00 | 0.00 | 1.00 | 1.00 | 1.00 | 1.00 | 1.00 | 1.00 | 0.00 | 1.00 | 1.00 | 1.00 | 1.00 | 1.00 | 1.00 |
| $A_5$ | 1.00 | 0.00 | 1.00 | 1.00 | 1.00 | 1.00 | 1.00 | 1.00 | 0.00 | 1.00 | 1.00 | 1.00 | 1.00 | 1.00 | 1.00 |
| $A_6$ | 1.00 | 0.00 | 1.00 | 1.00 | 1.00 | 1.00 | 0.00 | 1.00 | 0.00 | 1.00 | 1.00 | 1.00 | 1.00 | 1.00 | 1.00 |
| $A_7$ | 1.00 | 0.00 | 1.00 | 1.00 | 1.00 | 1.00 | 1.00 | 1.00 | 0.00 | 1.00 | 1.00 | 1.00 | 1.00 | 1.00 | 1.00 |
| $A_8$ | 1.00 | 0.00 | 1.00 | 1.00 | 1.00 | 0.00 | 0.00 | 1.00 | 0.00 | 1.00 | 1.00 | 1.00 | 1.00 | 1.00 | 1.00 |
| $A_9$ | 1.00 | 0.00 | 1.00 | 1.00 | 1.00 | 1.00 | 1.00 | 1.00 | 0.00 | 1.00 | 1.00 | 1.00 | 1.00 | 1.00 | 1.00 |
| $A_{10}$ | 1.00 | 0.00 | 1.00 | 1.00 | 1.00 | 1.00 | 1.00 | 1.00 | 0.00 | 1.00 | 1.00 | 1.00 | 1.00 | 1.00 | 1.00 |

**Table 18. Elderly people's satisfaction with the qualifications of service personnel.**

| $A_i$ | $p_{i1}^7$ | $p_{i2}^7$ | $p_{i3}^7$ | $p_{i4}^7$ | $p_{i5}^7$ | $p_{i6}^7$ | $p_{i7}^7$ | $p_{i8}^7$ | $p_{i9}^7$ | $p_{i10}^7$ | $p_{i11}^7$ | $p_{i12}^7$ | $p_{i13}^7$ | $p_{i14}^7$ | $p_{i15}^7$ |
|---|---|---|---|---|---|---|---|---|---|---|---|---|---|---|---|
| $A_1$ | 1.00 | 1.00 | 1.00 | 1.00 | 1.00 | 1.00 | 1.00 | 1.00 | 1.00 | 1.00 | 1.00 | 0.00 | 1.00 | 1.00 | 1.00 |
| $A_2$ | 0.00 | 0.00 | 0.00 | 0.00 | 0.00 | 0.00 | 1.00 | 0.00 | 0.00 | 0.00 | 0.00 | 0.00 | 0.00 | 0.00 | 0.00 |
| $A_3$ | 0.00 | 1.00 | 0.00 | 0.00 | 1.00 | 1.00 | 1.00 | 0.00 | 0.00 | 1.00 | 0.00 | 0.00 | 0.00 | 1.00 | 0.00 |
| $A_4$ | 0.00 | 0.00 | 1.00 | 0.00 | 1.00 | 1.00 | 1.00 | 1.00 | 0.00 | 1.00 | 1.00 | 0.00 | 0.00 | 1.00 | 0.00 |
| $A_5$ | 0.00 | 0.00 | 0.00 | 0.00 | 1.00 | 1.00 | 1.00 | 0.00 | 0.00 | 1.00 | 0.00 | 0.00 | 0.00 | 1.00 | 0.00 |
| $A_6$ | 0.00 | 0.00 | 0.00 | 0.00 | 0.00 | 1.00 | 1.00 | 0.00 | 0.00 | 1.00 | 0.00 | 0.00 | 0.00 | 0.00 | 0.00 |
| $A_7$ | 1.00 | 1.00 | 1.00 | 1.00 | 1.00 | 1.00 | 1.00 | 1.00 | 1.00 | 1.00 | 1.00 | 1.00 | 1.00 | 1.00 | 1.00 |
| $A_8$ | 0.00 | 0.00 | 0.00 | 0.00 | 0.00 | 0.00 | 0.00 | 0.00 | 0.00 | 0.00 | 0.00 | 0.00 | 0.00 | 0.00 | 0.00 |
| $A_9$ | 0.00 | 1.00 | 0.00 | 0.00 | 1.00 | 1.00 | 1.00 | 0.00 | 0.00 | 1.00 | 0.00 | 0.00 | 0.00 | 1.00 | 0.00 |
| $A_{10}$ | 0.00 | 0.00 | 0.00 | 0.00 | 0.00 | 1.00 | 1.00 | 0.00 | 0.00 | 1.00 | 0.00 | 0.00 | 0.00 | 0.00 | 0.00 |

Table 19 displays the relative importance that senior citizens place on each service personnel indicator. Table 20 displays the formula used to determine the elderly population's overall satisfaction with the service personnel.

Let $f_3^j = \begin{bmatrix} h_1^j & h_2^j \end{bmatrix}$ denote the indicator matrix of the $j$th service personnel's value for the indicator $E_3$ when selecting the elderly, and $g_3^i = \begin{bmatrix} h_1^i & h_2^i \end{bmatrix}$ denote the indicator matrix of the $i$th elderly's actual value for the indicator $E_3$, $r = 1,2$, $h_1^i + h_2^i = 1$. Let $q_{ij}^3$ denote the satisfaction of the $j$th service personnel with the $i$th elder person's work form indicator $E_3$. The formula for $q_{ij}^3$ is shown in Eq (20), and the results of the calculation are shown in Table 21.

$$q_{ij}^3 = \begin{cases} 1, f_3^i \geq g_3^j \\ 0, f_3^i < g_3^j \end{cases} \tag{20}$$

Let $f_4^j = \begin{bmatrix} k_1^j & k_2^j & k_3^j & k_4^j & k_5^j & k_6^j \end{bmatrix}$ denote the indicator matrix of the $j$th service personnel's values for indicator $E_4$ when selecting the elderly, and $g_4^i = \begin{bmatrix} k_1^i & k_2^i & k_3^i & k_4^i & k_5^i & k_6^i \end{bmatrix}$ denote the actual indicator matrix of the $i$th elderly person's values for the elderly situation indicator $E_4$, $k_v^j, k_v^i \in \{0, 1\}$, $v = 1,2,\ldots,6$, $\sum_{v=1}^{6} k_v^i = 1$. Denote $q_{ij}^4$ to denote the satisfaction of the $j$th service worker for the $i$th elderly person's elderly situation indicator $E_4$. The formula

**Table 19. The degree to which the elderly value various indicators.**

| $A_i$ | $w_i^1$ | $w_i^2$ | $w_i^3$ | $w_i^4$ | $w_i^5$ | $w_i^6$ | $w_i^7$ |
|---|---|---|---|---|---|---|---|
| $A_1$ | 0.08 | 0.04 | 0.04 | 0.04 | 0.15 | 0.25 | 0.4 |
| $A_2$ | 0.08 | 0.04 | 0.04 | 0.04 | 0.2 | 0.2 | 0.4 |
| $A_3$ | 0.06 | 0.06 | 0.04 | 0.04 | 0.05 | 0.2 | 0.55 |
| $A_4$ | 0.06 | 0.06 | 0.04 | 0.04 | 0.25 | 0.15 | 0.4 |
| $A_5$ | 0.08 | 0.04 | 0.04 | 0.04 | 0.1 | 0.2 | 0.5 |
| $A_6$ | 0.06 | 0.06 | 0.04 | 0.04 | 0.2 | 0.2 | 0.4 |
| $A_7$ | 0.08 | 0.04 | 0.04 | 0.04 | 0.05 | 0.25 | 0.5 |
| $A_8$ | 0.08 | 0.04 | 0.04 | 0.04 | 0.3 | 0.2 | 0.3 |
| $A_9$ | 0.08 | 0.04 | 0.04 | 0.04 | 0.1 | 0.2 | 0.5 |
| $A_{10}$ | 0.06 | 0.06 | 0.04 | 0.04 | 0.05 | 0.15 | 0.6 |

**Table 20. The total satisfaction of the elderly to the service personnel.**

| $A_i$ | $p_{i1}$ | $p_{i2}$ | $p_{i3}$ | $p_{i4}$ | $p_{i5}$ | $p_{i6}$ | $p_{i7}$ | $p_{i8}$ | $p_{i9}$ | $p_{i10}$ | $p_{i11}$ | $p_{i12}$ | $p_{i13}$ | $p_{i14}$ | $p_{i15}$ |
|---|---|---|---|---|---|---|---|---|---|---|---|---|---|---|---|
| $A_1$ | 0.92 | 0.73 | 0.99 | 1.00 | 0.94 | 0.92 | 0.84 | 0.81 | 0.66 | 0.93 | 0.90 | 0.41 | 0.98 | 0.81 | 0.80 |
| $A_2$ | 0.38 | 0.19 | 0.40 | 0.38 | 0.37 | 0.16 | 0.96 | 0.58 | 0.12 | 0.39 | 0.33 | 0.55 | 0.37 | 0.58 | 0.54 |
| $A_3$ | 0.34 | 0.75 | 0.42 | 0.43 | 0.90 | 0.88 | 0.99 | 0.37 | 0.18 | 0.90 | 0.31 | 0.44 | 0.41 | 0.93 | 0.43 |
| $A_4$ | 0.52 | 0.43 | 1.00 | 0.60 | 0.93 | 0.91 | 0.75 | 0.69 | 0.39 | 0.94 | 0.88 | 0.34 | 0.58 | 0.71 | 0.33 |
| $A_5$ | 0.43 | 0.30 | 0.48 | 0.50 | 0.93 | 0.91 | 0.90 | 0.37 | 0.22 | 0.93 | 0.39 | 0.36 | 0.50 | 0.88 | 0.37 |
| $A_6$ | 0.55 | 0.38 | 0.60 | 0.58 | 0.55 | 0.93 | 0.57 | 0.34 | 0.34 | 0.96 | 0.52 | 0.36 | 0.56 | 0.36 | 0.36 |
| $A_7$ | 0.89 | 0.69 | 0.94 | 0.98 | 0.87 | 0.87 | 0.99 | 0.93 | 0.74 | 0.89 | 0.86 | 0.97 | 0.95 | 0.93 | 0.98 |
| $A_8$ | 0.39 | 0.17 | 0.37 | 0.36 | 0.38 | 0.19 | 0.45 | 0.67 | 0.12 | 0.39 | 0.40 | 0.64 | 0.35 | 0.67 | 0.64 |
| $A_9$ | 0.40 | 0.77 | 0.45 | 0.49 | 0.90 | 0.88 | 0.90 | 0.33 | 0.30 | 0.90 | 0.36 | 0.38 | 0.48 | 0.84 | 0.40 |
| $A_{10}$ | 0.39 | 0.24 | 0.39 | 0.37 | 0.37 | 0.97 | 0.97 | 0.37 | 0.19 | 1.00 | 0.36 | 0.36 | 0.37 | 0.39 | 0.36 |

for q is shown in Eq (21), and the result of the calculation is shown in Table 22.

$$q_{ij}^4 = \begin{cases} 1, f_4^i \geq g_4^j \\ 0, f_4^i < g_4^j \end{cases} \tag{21}$$

Table 23 displays the level of significance that the service staff gave to each elderly indicator. The formula presented in Table 24 was utilized to determine the overall level of satisfaction that senior citizens had with the service staff.

Let $x_{ij}$ be a 0–1 type decision variable, $x_{ij} = 1$ means that the elderly $A_i$ and the service personnel $B_j$ form a match; otherwise $x_{ij} = 0$. Based on the satisfaction $p_{ij}$ of the elderly to the service personnel and the satisfaction $q_{ij}$ of the service personnel to the elderly, we can construct the following bilateral matching optimization model (22) with the objective of maximizing the satisfaction of the elderly and the satisfaction of the service personnel.

$$\max Z_1 = \sum_{i=1}^{m} \sum_{j=1}^{n} p_{ij} x_{ij} \tag{22A}$$

**Table 21. Satisfaction of service personnel with the form of work with elder persons.**

| $B_j$ | $q_{1j}^3$ | $q_{2j}^3$ | $q_{3j}^3$ | $q_{4j}^3$ | $q_{5j}^3$ | $q_{6j}^3$ | $q_{7j}^3$ | $q_{8j}^3$ | $q_{9j}^3$ | $q_{10j}^3$ |
|---|---|---|---|---|---|---|---|---|---|---|
| $B_1$ | 1.00 | 1.00 | 1.00 | 1.00 | 1.00 | 1.00 | 1.00 | 1.00 | 1.00 | 1.00 |
| $B_2$ | 1.00 | 1.00 | 1.00 | 1.00 | 1.00 | 1.00 | 1.00 | 1.00 | 1.00 | 1.00 |
| $B_3$ | 1.00 | 0.00 | 1.00 | 1.00 | 1.00 | 1.00 | 1.00 | 1.00 | 0.00 | 1.00 |
| $B_4$ | 1.00 | 1.00 | 1.00 | 1.00 | 1.00 | 1.00 | 1.00 | 1.00 | 1.00 | 1.00 |
| $B_5$ | 1.00 | 0.00 | 1.00 | 1.00 | 1.00 | 1.00 | 1.00 | 1.00 | 0.00 | 1.00 |
| $B_6$ | 1.00 | 1.00 | 1.00 | 1.00 | 1.00 | 1.00 | 1.00 | 1.00 | 1.00 | 1.00 |
| $B_7$ | 1.00 | 0.00 | 1.00 | 1.00 | 1.00 | 1.00 | 1.00 | 1.00 | 0.00 | 1.00 |
| $B_8$ | 1.00 | 1.00 | 1.00 | 1.00 | 1.00 | 1.00 | 1.00 | 1.00 | 1.00 | 1.00 |
| $B_9$ | 1.00 | 0.00 | 1.00 | 1.00 | 1.00 | 1.00 | 1.00 | 1.00 | 0.00 | 1.00 |
| $B_{10}$ | 0.00 | 1.00 | 0.00 | 0.00 | 0.00 | 0.00 | 0.00 | 0.00 | 1.00 | 0.00 |
| $B_{11}$ | 1.00 | 1.00 | 1.00 | 1.00 | 1.00 | 1.00 | 1.00 | 1.00 | 1.00 | 1.00 |
| $B_{12}$ | 1.00 | 1.00 | 1.00 | 1.00 | 1.00 | 1.00 | 1.00 | 1.00 | 1.00 | 1.00 |
| $B_{13}$ | 1.00 | 1.00 | 1.00 | 1.00 | 1.00 | 1.00 | 1.00 | 1.00 | 1.00 | 1.00 |
| $B_{14}$ | 1.00 | 1.00 | 1.00 | 1.00 | 1.00 | 1.00 | 1.00 | 1.00 | 1.00 | 1.00 |
| $B_{15}$ | 1.00 | 1.00 | 1.00 | 1.00 | 1.00 | 1.00 | 1.00 | 1.00 | 1.00 | 1.00 |

**Table 22. Satisfaction of service personnel with the situation of the elderly.**

| $B_j$ | $q_{1j}^4$ | $q_{2j}^4$ | $q_{3j}^4$ | $q_{4j}^4$ | $q_{5j}^4$ | $q_{6j}^4$ | $q_{7j}^4$ | $q_{8j}^4$ | $q_{9j}^4$ | $q_{10j}^4$ |
|---|---|---|---|---|---|---|---|---|---|---|
| $B_1$ | 1.00 | 1.00 | 1.00 | 1.00 | 1.00 | 1.00 | 1.00 | 1.00 | 1.00 | 1.00 |
| $B_2$ | 1.00 | 1.00 | 1.00 | 1.00 | 1.00 | 1.00 | 1.00 | 1.00 | 0.00 | 1.00 |
| $B_3$ | 1.00 | 1.00 | 1.00 | 1.00 | 1.00 | 0.00 | 1.00 | 0.00 | 1.00 | 0.00 |
| $B_4$ | 1.00 | 1.00 | 1.00 | 1.00 | 1.00 | 1.00 | 1.00 | 1.00 | 0.00 | 1.00 |
| $B_5$ | 1.00 | 1.00 | 1.00 | 1.00 | 1.00 | 1.00 | 1.00 | 1.00 | 1.00 | 1.00 |
| $B_6$ | 1.00 | 1.00 | 1.00 | 1.00 | 1.00 | 1.00 | 1.00 | 1.00 | 1.00 | 1.00 |
| $B_7$ | 1.00 | 1.00 | 1.00 | 1.00 | 1.00 | 0.00 | 1.00 | 0.00 | 1.00 | 0.00 |
| $B_8$ | 1.00 | 1.00 | 1.00 | 1.00 | 1.00 | 1.00 | 1.00 | 1.00 | 1.00 | 1.00 |
| $B_9$ | 1.00 | 1.00 | 1.00 | 1.00 | 1.00 | 1.00 | 1.00 | 1.00 | 0.00 | 1.00 |
| $B_{10}$ | 1.00 | 1.00 | 1.00 | 1.00 | 1.00 | 1.00 | 1.00 | 1.00 | 1.00 | 1.00 |
| $B_{11}$ | 1.00 | 1.00 | 1.00 | 1.00 | 1.00 | 1.00 | 1.00 | 1.00 | 1.00 | 1.00 |
| $B_{12}$ | 1.00 | 1.00 | 1.00 | 1.00 | 1.00 | 0.00 | 1.00 | 0.00 | 1.00 | 0.00 |
| $B_{13}$ | 1.00 | 1.00 | 1.00 | 1.00 | 1.00 | 1.00 | 1.00 | 1.00 | 1.00 | 1.00 |
| $B_{14}$ | 1.00 | 1.00 | 1.00 | 1.00 | 1.00 | 1.00 | 1.00 | 1.00 | 1.00 | 1.00 |
| $B_{15}$ | 1.00 | 1.00 | 1.00 | 1.00 | 1.00 | 1.00 | 1.00 | 1.00 | 1.00 | 1.00 |

$$\max Z_2 = \sum_{i=1}^m \sum_{j=1}^n q_{ij} x_{ij} \tag{22B}$$

$$s.t. \sum_{j=1}^n x_{ij} \le 1 \tag{22C}$$

$$\sum_{i=1}^m x_{ij} \le 1 \tag{22D}$$

**Table 23. Level of importance attached by service personnel to each indicator.**

| $B_j$ | $w_j^1$ | $w_j^2$ | $w_j^3$ | $w_j^4$ |
|---|---|---|---|---|
| $B_1$ | 0.48 | 0.32 | 0.1 | 0.1 |
| $B_2$ | 0.48 | 0.32 | 0.15 | 0.05 |
| $B_3$ | 0.48 | 0.32 | 0.05 | 0.15 |
| $B_4$ | 0.48 | 0.32 | 0.1 | 0.1 |
| $B_5$ | 0.56 | 0.24 | 0.1 | 0.1 |
| $B_6$ | 0.56 | 0.24 | 0.15 | 0.05 |
| $B_7$ | 0.56 | 0.24 | 0.15 | 0.05 |
| $B_8$ | 0.56 | 0.24 | 0.1 | 0.1 |
| $B_9$ | 0.56 | 0.24 | 0.05 | 0.15 |
| $B_{10}$ | 0.56 | 0.24 | 0.1 | 0.1 |
| $B_{11}$ | 0.56 | 0.24 | 0.1 | 0.1 |
| $B_{12}$ | 0.48 | 0.32 | 0.15 | 0.05 |
| $B_{13}$ | 0.48 | 0.32 | 0.1 | 0.1 |
| $B_{14}$ | 0.48 | 0.32 | 0.15 | 0.05 |
| $B_{15}$ | 0.48 | 0.32 | 0.1 | 0.1 |

**Table 24. Overall satisfaction of service personnel with older persons.**

| $B_j$ | $q_{1j}$ | $q_{2j}$ | $q_{3j}$ | $q_{4j}$ | $q_{5j}$ | $q_{6j}$ | $q_{7j}$ | $q_{8j}$ | $q_{9j}$ | $q_{10j}$ |
|---|---|---|---|---|---|---|---|---|---|---|
| $B_1$ | 0.44 | 0.76 | 0.44 | 0.62 | 0.76 | 0.54 | 0.52 | 0.92 | 0.54 | 0.82 |
| $B_2$ | 0.63 | 0.94 | 0.60 | 0.86 | 1.00 | 0.79 | 0.66 | 0.94 | 0.68 | 0.87 |
| $B_3$ | 0.94 | 0.89 | 0.88 | 1.00 | 1.00 | 0.85 | 0.88 | 0.79 | 0.82 | 0.72 |
| $B_4$ | 0.94 | 0.94 | 0.81 | 1.00 | 1.00 | 1.00 | 0.81 | 0.94 | 0.77 | 0.87 |
| $B_5$ | 0.92 | 0.86 | 0.74 | 1.00 | 1.00 | 0.96 | 0.78 | 0.96 | 0.82 | 0.92 |
| $B_6$ | 0.64 | 0.96 | 0.57 | 0.83 | 1.00 | 0.79 | 0.61 | 0.96 | 0.75 | 0.92 |
| $B_7$ | 0.94 | 0.85 | 0.72 | 1.00 | 1.00 | 0.89 | 0.78 | 0.89 | 0.79 | 0.89 |
| $B_8$ | 0.38 | 0.38 | 0.44 | 0.44 | 0.44 | 0.44 | 0.44 | 0.78 | 0.38 | 0.56 |
| $B_9$ | 0.90 | 0.90 | 0.84 | 1.00 | 1.00 | 0.95 | 0.89 | 0.95 | 0.70 | 0.90 |
| $B_{10}$ | 0.40 | 0.83 | 0.29 | 0.52 | 0.73 | 0.56 | 0.34 | 0.85 | 0.62 | 0.85 |
| $B_{11}$ | 0.36 | 0.68 | 0.40 | 0.55 | 0.72 | 0.51 | 0.44 | 0.96 | 0.47 | 0.81 |
| $B_{12}$ | 0.92 | 1.00 | 0.82 | 1.00 | 1.00 | 0.87 | 0.90 | 0.87 | 0.92 | 0.87 |
| $B_{13}$ | 0.55 | 0.95 | 0.47 | 0.79 | 1.00 | 0.74 | 0.52 | 0.95 | 0.69 | 0.89 |
| $B_{14}$ | 0.55 | 0.86 | 0.46 | 0.65 | 0.86 | 0.71 | 0.52 | 0.94 | 0.65 | 0.94 |
| $B_{15}$ | 0.76 | 0.95 | 0.67 | 1.00 | 1.00 | 0.95 | 0.73 | 0.95 | 0.89 | 0.89 |

$$i = 1, 2, \ldots, m; j = 1, 2, \ldots, n \tag{22E}$$

In model (22), Eq (22A) and Eq (22B) are the objective functions, and Eq (22A) indicates that the elder's satisfaction with the service personnel is maximized; Eq (22B) indicates that the service personnel's satisfaction with the elder is maximized. Constraint (22c) means that each elder must be matched with one service personnel only and only one service personnel; and Constraint (22d) means that each service worker can only be matched with one elder person or not matched with an elder person.

Model (22) is a bi-objective 0–1 integer programming model that can be solved using the linear weighting method [64], the basic idea of which is to transform a multi-objective problem into an algorithm for single-objective planning by summing the objective functions in a linearly weighted manner. Model (23) is transformed into the following single-objective optimization model:

$$\max Z = \omega_1 \sum_{i=1}^{m} \sum_{j=1}^{n} p_{ij} x_{ij} + \omega_2 \sum_{i=1}^{m} \sum_{j=1}^{n} q_{ij} x_{ij} \tag{23A}$$

$$s.t. \sum_{j=1}^{n} x_{ij} \leq 1 \tag{23C}$$

$$\sum_{i=1}^{m} x_{ij} \leq 1 \tag{23D}$$

$$i = 1, 2, \ldots, m; j = 1, 2, \ldots, n \tag{23E}$$

The weights $\omega_1$ and $\omega_2$ reflect the importance of both subjects in the actual matching decision. If $\omega_1 > \omega_2$, it means that the satisfaction degree of the subject on the elderly side should be emphasized in the matching decision; if $\omega_1 < \omega_2$, it means that the satisfaction degree of the

subject on the service personnel side should be emphasized in the matching decision; and if $\omega_1 = \omega_2 = 0.5$, it means that the fairness of both subjects should be considered in the matching decision.

Then the optimization software LINGO 18.0 is used to solve the problem, and the matching results obtained are: $x_{1,7} = 1$, $x_{2,12} = 1$, $x_{3,9} = 1$, $x_{4,3} = 1$, $x_{5,4} = 1$, $x_{6,6} = 1$, $x_{7,15} = 1$, $x_{8,14} = 1$, $x_{9,13} = 1$, $x_{10,5} = 1$, i.e. $A_1 \leftrightarrow B_7$, $A_2 \leftrightarrow B_{12}$, $A_3 \leftrightarrow B_9$, $A_4 \leftrightarrow B_3$, $A_5 \leftrightarrow B_4$, $A_6 \leftrightarrow B_6$, $A_7 \leftrightarrow B_{15}$, $A_8 \leftrightarrow B_{14}$, $A_9 \leftrightarrow B_{13}$, $A_{10} \leftrightarrow B_5$. From the results, the traditional bilateral matching method to maximize satisfaction as the goal, the original hard indicators into satisfaction and lead to the results of the calculations as much as possible for the elderly to match the last service personnel so that their overall satisfaction is maximized, but from the results of the matching there may be a waste of resources or mutual requirements are not satisfied with the situation. For example, there is a waste of service resources in the $A_1 \leftrightarrow B_7$ matches and the elderly do not fully meet the requirements of the service personnel; $A_2 \leftrightarrow B_{12}$, $A_3 \leftrightarrow B_9$, $A_4 \leftrightarrow B_3$, $A_7 \leftrightarrow B_{15}$, $A_9 \leftrightarrow B_{13}$, $A_{10} \leftrightarrow B_5$ matches have a slight gap between the service resources needed by the elderly and the service resources are not quite fulfilled; and $A_8 \leftrightarrow B_{14}$ matches belong to the situation where the resources seriously do not meet the needs of the elderly. Although the elderly are matched with service personnel, the specific needs of the elderly are not always met.

## 7. Conclusion

In order to address the issue of elderly service personnel matching based on the smart elderly platform, we propose in this paper a matching method for elderly service personnel that takes into account multiple types of service expectations. The senior care service personnel matching model is constructed and solved in a targeted manner in order to determine the senior care service personnel matching program. This method fully considers the two typical scenarios of sufficient and insufficient senior care service personnel, as well as the expectations of the elderly and the senior care service personnel for each other. When the suggested methods are finally compared, the findings indicate that the suggested approach is more logical and useful. In order to improve the matching efficiency and effect, we should balance the needs and priorities of the elderly, pay attention to the elderly's service as the center, understand the specific needs of the elderly, and better fit reality. The staff matching method based on multi-type service expectations proposed in this paper has a wide range of practical applications. It also provides a theoretical reference for solving the matching problem of elderly service personnel in reality.

It should be noted that the research presented in this paper has certain limitations. Specifically, the two-way matching satisfaction is only calculated using a subset of representative indicators. But in actuality, there are a lot of variables that influence the bilateral matching between the elderly and elderly service personnel because of the intricacy of the market and the variety of matching indicators used in the intelligent pension platform's operation process. In order to examine the impact of additional behavioral factors on the outcomes of person-to-person matching, we can take into account additional behavioral factors in the next research project, such as psychological and behavioral factors, during the matching process between the elderly and elderly service personnel.

## Author Contributions

**Conceptualization:** Chao Yu, Tianxiang Gao.

**Data curation:** Chao Yu, Tianxiang Gao.

**Formal analysis:** Chao Yu, Tianxiang Gao.

**Funding acquisition:** Chao Yu.

**Investigation:** Chao Yu, Tianxiang Gao.

**Methodology:** Chao Yu, Tianxiang Gao.

**Project administration:** Chao Yu, Tianxiang Gao.

**Resources:** Chao Yu, Tianxiang Gao.

**Software:** Tianxiang Gao.

**Supervision:** Chao Yu.

**Validation:** Tianxiang Gao.

**Visualization:** Chao Yu, Tianxiang Gao.

**Writing – original draft:** Chao Yu, Tianxiang Gao.

**Writing – review & editing:** Chao Yu, Tianxiang Gao.

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
