## [Decision Letter · Decision Letter 0]

10 Jun 2024

PONE-D-24-19546A matching method for elderly care service personnel with multiple types of service expectationsPLOS ONE

Dear Dr. Gao,

Thank you for submitting your manuscript to PLOS ONE. After careful consideration, we feel that it has merit but does not fully meet PLOS ONE’s publication criteria as it currently stands. Therefore, we invite you to submit a revised version of the manuscript that addresses the points raised during the review process.

The manuscript has received mixed reviews from the reviewers, however, both of them agree that after careful restructuring, the manuscript may be refined for acceptance. Please carefully go through the reviewers comments and revise.

We look forward to receiving your revised manuscript.

Kind regards,

Yogesh Kumar Jain, MPH

Academic Editor

PLOS ONE

Journal Requirements:

4. Please ensure that you include a title page within your main document. You should list all authors and all affiliations as per our author instructions and clearly indicate the corresponding author.

Reviewers' comments:

Reviewer's Responses to Questions

**Comments to the Author**

1. Is the manuscript technically sound, and do the data support the conclusions?

Reviewer #1: Yes

Reviewer #2: Yes

2. Has the statistical analysis been performed appropriately and rigorously? 

Reviewer #1: Yes

Reviewer #2: N/A

3. Have the authors made all data underlying the findings in their manuscript fully available?

Reviewer #1: Yes

Reviewer #2: Yes

4. Is the manuscript presented in an intelligible fashion and written in standard English?

Reviewer #1: Yes

Reviewer #2: Yes

5. Review Comments to the Author

Reviewer #1: Strengths :

The article clearly identifies the problems of providing care to the elderly and expectations of the service personnel from the elderly.

The methodology describes in detail how to compute expectation indices and satisfaction levels of both elderly and service personnel.

The model developed for maximizing the satisfaction levels meets the objectives and the model is adaptable to two different scenarios (sufficient and insufficient personnel) making it feasible to use under both settings.

The statistics to support the results are appropriate and quite comprehensive.

Areas for improvement:

The article, especially the abstract, repeats information, regarding the goals and outcomes, which could be presented in a more condensed manner.

A few terms, such as elastic service expectation type and inelastic service expectation type, could have been introduced using a definition for better understanding of the concept.

Reviewer #2: The paper proposed a matching method for elderly care service personnel considering multi-type service expectations with a bilateral matching optimization model. The topic is of great interest and reality, however, the academic contribution should be refined and highlighted. Here is some advice.

1.The introduction is too lengthy and should be reduced, especially focused on the home-based service matching problem for elderly care.

2."It is evident that the matching issue faced by home care providers differs from the conventional matching issue in domestic services...." Why to assert this? What are the reasons? There are some other similar claims without reasons or references, eg., "According to a survey conducted by a few domestic help companies, elderly people typically have expectations related to their age when selecting elderly care service personnel.", "...and they typically select those with higher educational backgrounds...".

3.Some expressions are redundant or repeated, eg., "Additionally, there may be an imbalance between the supply and demand of elderly service resources in actual life.", "It should be emphasized that there may be an imbalance between the demand and supply for senior service persons in real life,...".

4.The main contribution and innovation should be presented at the end of Introduction section.

5.Section 2.2, papers of matching for elderly care should be supplemented. There are some special issues on this topic recent years, such as "Smart Technology-Supported Independent Living for Older Adults" in International Journal of Human-Computer Interaction in 2023.

6.What are the differences between heavy and light black lines in Fig. 1?

7.“...elder people have inelastic expectations about gender indicators.” It is very important to give the reason for this claim, since it's the basis to classify "Gender" into the group of inelastic expectations.

8.A comparison with other current matching methods under the same circumstances should be given to validate the outperformance of the proposed method.

9.Future research should be presented in Conclusion.

6. PLOS authors have the option to publish the peer review history of their article (what does this mean?). If published, this will include your full peer review and any attached files.

Reviewer #1: **Yes: **Rupali Gupta

Reviewer #2: No

---

## [Author Response · Author response to Decision Letter 0]

28 Jul 2024

We thank the referee very much for the comments and suggestions. They are very helpful for us to revise and improve the paper. The paper has been carefully revised according to the referee’s advice. We have made the following changes on the paper accordingly.

Reviewer #1, Concern #1: The article, especially the abstract, repeats information, regarding the goals and outcomes, which could be presented in a more condensed manner.

Author response: Thank you for your valuable feedback and for taking the time to review our manuscript. We sincerely appreciate your thorough evaluation, which has provided us with valuable insights to improve the quality of our work.

As suggested by the referee, the summary section has been truncated with information on repeated objectives and results.

The changes have been marked in revision mode, and the revised content is also indicated in red in the manuscript, which we have attached for your convenience.________________________________________

Reviewer #1, Concern #2: A few terms, such as elastic service expectation type and inelastic service expectation type, could have been introduced using a definition for better understanding of the concept.

Author response: Thank you very much for reviewing our paper and providing valuable feedback. We sincerely appreciate your thorough evaluation, which has provided us with valuable insights to improve the quality of our work.

According to referee’s comments, we added definitions of elastic and inelastic service expectation types in the “Problem description” section to better understand this concept.

The changes have been marked in revision mode, and the revised content is also indicated in red in the manuscript, which we have attached for your convenience.

Reviewer #2, Concern #1: The introduction is too lengthy and should be reduced, especially focused on the home-based service matching problem for elderly care.

Author response: Thank you for your valuable feedback and for taking the time to review our manuscript. We sincerely appreciate your thorough evaluation, which has provided us with valuable insights to improve the quality of our work.

According to referee’s comments, we have already made appropriate cuts in the “Introduction” section.

The changes have been marked in revision mode, and the revised content is also indicated in red in the manuscript, which we have attached for your convenience.

Reviewer #2, Concern #2: "It is evident that the matching issue faced by home care providers differs from the conventional matching issue in domestic services...." Why to assert this? What are the reasons? There are some other similar claims without reasons or references, eg., "According to a survey conducted by a few domestic help companies, elderly people typically have expectations related to their age when selecting elderly care service personnel.", "...and they typically select those with higher educational backgrounds...".

Author response: Thank you for reviewing our manuscript and providing valuable feedback. We sincerely appreciate your thorough evaluation, which has provided us with valuable insights to improve the quality of our work.

As suggested by the reviewer, in the description of the indicators, we added references as a basis for consideration.

[1] Blažienė I, Žalimienė L, Between user’s expectations and provider’s quality of work: The future of elderly care in Lithuania, Journal of Population Ageing, 13 (2020), 5-23. https://doi.org/10.10 07/s12062-017-9215-1.

[2] Burch K A, Dugan A G, Barnes-Farrell J L, Understanding what eldercare means for employees and organizations: a review and recommendations for future research, Work, Aging and Retirement, 5 (2019), 44-72. https://doi.org/10.1093/workar/way011.

[3] Carpenter B D, Van Haitsma K, Ruckdeschel K, et al, The psychosocial preferences of older adults: a pilot examination of content and structure1, The Gerontologist, 40 (20000), 335-348. https://doi.org/10.1093/geront/40.3.335.

[4] Maiden R J, Horowitz B P, Howe J L, Workforce training and education gaps in gerontology and geriatrics: what we found in New York State, Gerontology & Geriatrics Education, 31 (2021), 328-348. https://doi.org/10.1080/02701960.2010.532749.

The changes have been marked in revision mode, and the revised content is also indicated in red in the manuscript, which we have attached for your convenience.

Reviewer #2, Concern #3: Some expressions are redundant or repeated, eg., "Additionally, there may be an imbalance between the supply and demand of elderly service resources in actual life.", "It should be emphasized that there may be an imbalance between the demand and supply for senior service persons in real life,...".

Author response: Thank you for your valuable feedback and for taking the time to review our manuscript. We sincerely appreciate your thorough evaluation, which has provided us with valuable insights to improve the quality of our work.

As suggested by the reviewer, we have cut down on expressions that are redundant or repetitive.

The changes have been marked in revision mode, and the revised content is also indicated in red in the manuscript, which we have attached for your convenience. 

Reviewer #2, Concern #4: The main contribution and innovation should be presented at the end of Introduction section.

Author response: Thank you very much for reviewing our paper and providing valuable feedback. We sincerely appreciate your thorough evaluation, which has provided us with valuable insights to improve the quality of our work.

 In response to the reviewers' feedback, we add the main points of innovation at the end of the “Introduction” section.

The changes have been marked in revision mode, and the revised content is also indicated in red in the manuscript, which we have attached for your convenience.

Reviewer #2, Concern #5: Section 2.2, papers of matching for elderly care should be supplemented. There are some special issues on this topic recent years, such as "Smart Technology-Supported Independent Living for Older Adults" in International Journal of Human-Computer Interaction in 2023.

Author response: Thank you very much for reviewing our paper and providing valuable feedback. We sincerely appreciate your thorough evaluation, which has provided us with valuable insights to improve the quality of our work.

As suggested by the referee, we added in Section 2.2 four pieces of literature on matching documents for elderly care and some particular issues that have arisen in recent years on this topic. [1] Zhou J, Smart technology-supported independent living for older adults: an editorial, International Journal of Human–Computer Interaction, 39 (2023), 961-963. https://doi.org/10.1080/10447318.2023.2170518.

[2] Liu N, Pu Q, Shi Y, et al, Older adults’ interaction with intelligent virtual assistants: the role of information modality and feedback, International Journal of Human–Computer Interaction, 39 (2023), 1162-1183. https://doi.org/10.1080/10447318.202 2.2074667.

[3] Wen P, Chen M, A new model for elderly emotional care routing and scheduling with multi-agency and the combination of nearby services, International Journal of Human–Computer Interaction, 39 (2023), 1111-1120. https://doi.org/10.1080/10447318.202 2.2050544.

[4] Sun X, Ding J, Dong Y, et al, A survey of technologies facilitating home and community-based stroke rehabilitation, International Journal of Human–Computer Interaction, 39 (2023), 1016-1042. https://doi.org/10.1080/10447318.2022.2050545.The changes have been marked in revision mode, and the revised content is also indicated in red in the manuscript, which we have attached for your convenience.________________________________________

Reviewer #2, Concern #6: What are the differences between heavy and light black lines in Fig. 1?

Author response: Thank you very much for reviewing our paper and providing valuable feedback. We sincerely appreciate your thorough evaluation, which has provided us with valuable insights to improve the quality of our work.

In response to the reviewers' feedback, we've added some descriptions of figure 1 in the "Problem description" section. In figure 1, the thin arrow indicates that both the elderly and the service provider are satisfied with each other; the thick arrow indicates that both are matched, that is, satisfy each other's requirements and satisfy the highest overall satisfaction.

The changes have been marked in revision mode, and the revised content is also indicated in red in the manuscript, which we have attached for your convenience.

Reviewer #2, Concern #7: “...elder people have inelastic expectations about gender indicators.” It is very important to give the reason for this claim, since it's the basis to classify "Gender" into the group of inelastic expectations.

Author response: Thank you for your valuable feedback and for taking the time to review our manuscript. We sincerely appreciate your thorough evaluation, which has provided us with valuable insights to improve the quality of our work.

According to referee’s comments, our reasoning for this assertion is based on online and offline survey statistics, with additional references.

[1] Zuo M. Intelligent service and operation for the aged, Beijing: Tsinghua University Press, 2022. ISBN：9787302595328.

The changes have been marked in revision mode, and the revised content is also indicated in red in the manuscript, which we have attached for your convenience.

Reviewer #2, Concern #8: A comparison with other current matching methods under the same circumstances should be given to validate the outperformance of the proposed method.

Author response: Thank you for your valuable feedback and for taking the time to review our manuscript. We sincerely appreciate your thorough evaluation, which has provided us with valuable insights to improve the quality of our work.

According to referee’s comments, the section "6. Comparison" has been incorporated into the body of the paper. When using the conventional two-sided matching method, neither side's needs can be satisfied nor service resources will be wasted. The suggested approach is more focused, better able to represent the unique needs of the elderly, and places more emphasis on the elderly-centered service concept than the conventional two-sided matching method.

The changes have been marked in revision mode, and the revised content is also indicated in red in the manuscript, which we have attached for your convenience.

Reviewer #2, Concern #9: Future research should be presented in Conclusion.

Author response: Thank you for your valuable feedback and for taking the time to review our manuscript. We sincerely appreciate your thorough evaluation, which has provided us with valuable insights to improve the quality of our work.

According to referee’s comments, in the conclusions section, we have included a discussion of the study's limitations and future directions. In order to examine the impact of additional behavioral factors on the outcomes of person matching, we will take into account psychological and behavioral factors in addition to the behavioral ones when matching senior citizens with service providers.

The changes have been marked in revision mode, and the revised content is also indicated in red in the manuscript, which we have attached for your convenience.

---

## [Decision Letter · Decision Letter 1]

13 Aug 2024

A matching method for elderly care service personnel with multiple types of service expectations

PONE-D-24-19546R1

Dear Dr. Gao,

We’re pleased to inform you that your manuscript has been judged scientifically suitable for publication and will be formally accepted for publication once it meets all outstanding technical requirements.

Kind regards,

Yogesh Kumar Jain, MPH

Academic Editor

PLOS ONE

Additional Editor Comments (optional):

Reviewers' comments:

Reviewer's Responses to Questions

**Comments to the Author**

1. If the authors have adequately addressed your comments raised in a previous round of review and you feel that this manuscript is now acceptable for publication, you may indicate that here to bypass the “Comments to the Author” section, enter your conflict of interest statement in the “Confidential to Editor” section, and submit your "Accept" recommendation.

Reviewer #2: All comments have been addressed

2. Is the manuscript technically sound, and do the data support the conclusions?

Reviewer #2: Yes

3. Has the statistical analysis been performed appropriately and rigorously? 

Reviewer #2: Yes

4. Have the authors made all data underlying the findings in their manuscript fully available?

Reviewer #2: Yes

5. Is the manuscript presented in an intelligible fashion and written in standard English?

Reviewer #2: Yes

6. Review Comments to the Author

Reviewer #2: The paper proposed a matching method for elderly care service personnel considering multi-type service expectations with a bilateral matching optimization model. The revised version is much better and well responds to my concerns. Good luck.

7. PLOS authors have the option to publish the peer review history of their article (what does this mean?). If published, this will include your full peer review and any attached files.

Reviewer #2: No

---

## [Editor Report · Acceptance letter]

14 Sep 2024

PONE-D-24-19546R1 

PLOS ONE

Dear Dr. Gao, 

I'm pleased to inform you that your manuscript has been deemed suitable for publication in PLOS ONE. Congratulations! Your manuscript is now being handed over to our production team.

Kind regards, 

on behalf of

Dr. Yogesh Kumar Jain 

Academic Editor

PLOS ONE